# Hepatic arginase 2 (Arg2) is sufficient to convey the therapeutic metabolic effects of fasting

Yiming Zhang[1], Cassandra B. Higgins[1], Hannah M. Fortune[1], Phillip Chen[1], Alicyn I. Stothard[2], Allyson L. Mayer [1], Benjamin M. Swarts[2] & Brian J. DeBosch[1,3]

Caloric restriction and intermittent fasting are emerging therapeutic strategies against obesity, insulin resistance and their complications. However, the effectors that drive this response are not completely defined. Here we identify arginase 2 (Arg2) as a fasting-induced hepatocyte factor that protects against hepatic and peripheral fat accumulation, hepatic inflammatory responses, and insulin and glucose intolerance in obese murine models. Arg2 is upregulated in fasting conditions and upon treatment with the hepatocyte glucose transporter inhibitor trehalose. Hepatocyte-specific Arg2 overexpression enhances basal thermogenesis, and protects from weight gain, insulin resistance, glucose intolerance, hepatic steatosis and hepatic inflammation in diabetic mouse models. Arg2 suppresses expression of the regulator of G-protein signalling (RGS) 16, and genetic RGS16 reconstitution reverses the effects of Arg2 overexpression. We conclude that hepatocyte Arg2 is a critical effector of the hepatic glucose fasting response and define a therapeutic target to mitigate the complications of obesity and non-alcoholic fatty liver disease.

[1] Department of Pediatrics, Washington University School of Medicine, St. Louis, MO 63110, USA. [2] Department of Chemistry & Biochemistry, Central Michigan University, Mt. Pleasant, MI 48859, USA. [3] Department of Cell Biology & Physiology, Washington University School of Medicine, St. Louis, MO 63110, USA. Correspondence and requests for materials should be addressed to B.J.D. (email: deboschb@wustl.edu)

O besigenic cardiometabolic disease arises from a constellation of environmental, genetic and meta- and epigenetic factors[1]. Clinically, intensive lifestyle management remains the primary means of treating obesity and its comorbidities, including insulin resistance, non-alcoholic fatty liver disease (NAFLD), and thermic depression. Intensive lifestyle management encompasses a wide range of interventions, such as increased locomotion, selective macronutrient elimination (e.g., ketogenic diets), intermittent fasting (IF), and caloric restriction (CR)[2]. Although lifestyle measures effectively treat obesity, insulin resistance and NAFLD[2–8], caloric restriction, fasting and extreme dietary alterations are often unsustainable, unpalatable, and are in some cases associated with clinical complications[9–11].

At the nexus of portal and systemic circulations, the hepatocyte coordinates peripheral responses to changes in nutrient flux. During physiological fasting, the hepatocyte provides glucose and ketones as fuel for fasting peripheral organs. In addition, hepatocyte-derived endocrine signals, such as fibroblast growth factor 21 (FGF21), enhance peripheral insulin sensitivity and promote basal thermogenesis. These adaptations ostensibly maintain thermic equilibrium during prolonged fasting and hibernation, and maximize post-fasting nutrient absorption[12–16]. The physiological fasting response is itself induced by several key transcriptional regulators that are also activated during fasting: peroxisome proliferator activated receptor α (PPARα), PPARγ coactivator 1α (PGC1α), SIRT1[17,18], and transcription factor EB (TFEB)[19–23]. However, surprisingly little is known regarding hepatocyte-intrinsic factors that regulate hepatic and extrahepatic responses to fasting.

We recently demonstrated that hepatic glucose transport blockade is sufficient to confer or augment the adaptive metabolic changes associated with generalized caloric restriction and intermittent fasting[24–29]. Germline deletion of the hepatic glucose transporter (GLUT) 8 increased hepatocyte fatty acid oxidation and decreased fructose-induced de novo lipogenesis in isolated hepatocytes[25]. In vivo, targeted GLUT8 disruption increased basal thermogenesis[25,27,28], protected from diet-induced NAFLD and insulin resistance, and augmented canonical hepatic fasting responses (e.g., PPARα and FGF21[25,27,28]). Similarly, pharmacological hepatic GLUT inhibition by the GLUT inhibitor, trehalose, activated hepatocyte fasting responses—autophagic flux and FGF21 secretion—that was dependent on hepatocyte TFEB[24,30,31]. The hepatic response to pharmacological hepatic GLUT inhibition is to activate thermogenesis, and reduce hepatic steatosis and insulin resistance[24,26,30,32]. The full mechanisms driving the hepatocyte glucose fasting response and the utility of these responses against cardiometabolic disease, however, are only now being elucidated.

Here, we identify the arginine metabolizing enzyme, arginase 2 (Arg2) as a hepatocyte glucose fasting-induced factor that is sufficient to convey the adaptive hepatic and peripheral metabolic changes induced during CR and IF. Hepatocyte *Arg2* mRNA and protein expression is induced during acute and prolonged fasting, and after treatment with the hepatic GLUT inhibitor, trehalose. Hepatocyte-specific Arg2 overexpression increases basal thermogenesis, enhances insulin sensitivity, and decreases hepatic steatosis and inflammation in genetic and diet-induced models of diabetes and NAFLD. Unbiased transcriptomic analysis identifies the regulator of G-protein-coupled receptor signaling (RGS) 16 to be among the most highly suppressed hepatic genes in Arg2-overexpressing *db/db* diabetic mice. Moreover, hepatocyte-specific genetic RGS16 reconstitution in the context of Arg2 overexpression reverses the hepatic and peripheral therapeutic effects of Arg2 overexpression. We conclude that Arg2 represents an effector of the hepatocyte-intrinsic fasting response that modulates hepatic and extrahepatic metabolic homeostasis, and

that the trehalose class of disaccharide GLUT inhibitors activates this pathway. More broadly, this study identifies a seminal link between hepatic Arg2 and G-protein signaling in the pathogenesis of metabolic disease.

## Results

**Arg2 is induced during fasting and pseudo-fasting.** We previously demonstrated that glucose transporter inhibition by the hepatic GLUT inhibitor, trehalose, enhances peripheral thermogenesis, and mitigates hepatic steatosis by inducing hepatocyte fasting-like pathways[24,26,29]. RNA-seq analysis of trehalose-treated primary hepatocytes identified hepatocyte arginase 2 (Arg2) to be among the most highly upregulated genes relative to untreated hepatocytes. We confirmed these data by quantitative real-time RT-PCR (qPCR) and immunoblot analysis, demonstrating rapid, dose-dependent, cell autonomous Arg2 induction within 24 h post trehalose (1 mM, 10 mM, and 100 mM) treatment (Fig. 1a). This Arg2 response was energy deficit-dependent, as indicated by pre-treating cells with pyruvate as an energy substrate to bypass the GLUTs targeted by 100 mM trehalose. Pyruvate significantly reduced the trehalose-induced Arg2 response to trehalose (Fig. 1b). Because trehalose catabolism by host (brush border, hepatocyte, or renal) or microbiotic trehalases can theoretically reduce the bioavailability of orally administered trehalose therapy, we compared the potency of native trehalose with that of a trehalase-resistant analog, lactotrehalose[32–34], to induce Arg2 expression in vitro (Fig. 1c). Lactotrehalose-induced Arg2 expression indeed was significantly enhanced at 10 mM and 100 mM when compared with native trehalose at equivalent doses (Fig. 1c). Trehalose similarly induced *Arg2* mRNA and protein in livers from mice fed 3% trehalose water (5 days) ad libitum in vivo (Fig. 1d, and Supplementary Fig. 6). The similarities between trehalose-induced pseudo-fasting responses in the liver and physiological macronutrient withdrawal prompted us to quantify the Arg2 response in livers from mice fasting up to 24 h. qPCR and immunoblot analysis identified Arg2 to be activated during physiological fasting (Fig. 1e, and Supplementary Fig. 6). Arg2 upregulation during fasting is cell-intrinsic, as determined by subjecting isolated primary murine hepatocytes to serum and glucose deprivation (24 h), after which Arg2 mRNA expression was significantly upregulated (Fig. 1f).

We demonstrated that hepatocyte glucose transport blockade attenuates insulin resistance and hepatic steatosis in multiple obese models[24,25,27–29]. To test the hypothesis that hepatocyte Arg2 activation downstream of the fasting response is sufficient to drive these therapeutic effects, we examined the cell-intrinsic consequences of forced Arg2 overexpression on hepatocyte triglyceride accumulation, inflammatory cytokine and chemokine activation, and insulin signal transduction. BSA-conjugated fatty acid treatment (FFA) both without and with 1 nM lipopolysaccharide (LPS) induced TG accumulation and inflammatory gene markers interleukin 1β (*Il-1β*), interleukin 6 (*Il-6*), tumor necrosis factor α (*Tnf-α*), monocyte chemoattractant protein 1 (*Mcp1*; also known as *Ccl2*), and CXC chemokine ligand 9 (*Cxcl9*) in wild-type primary hepatocyte cultures (Fig. 1g). In each condition, fat accumulation and cytokine/chemokine expression was significantly attenuated by Arg2 overexpression (Fig. 1g). In parallel, we verified expression of functional protein by quantifying urea production in vitro, which revealed an eightfold increase in urea production upon Arg2 overexpression (not shown). To define the effect of Arg2 overexpression on hepatocyte insulin sensitivity in vitro, primary hepatocytes were transfected with β-galactosidase or Arg2, serum-deprived overnight, then stimulated with 10 nM recombinant insulin. Arg2-overexpressing cultures exhibited enhanced

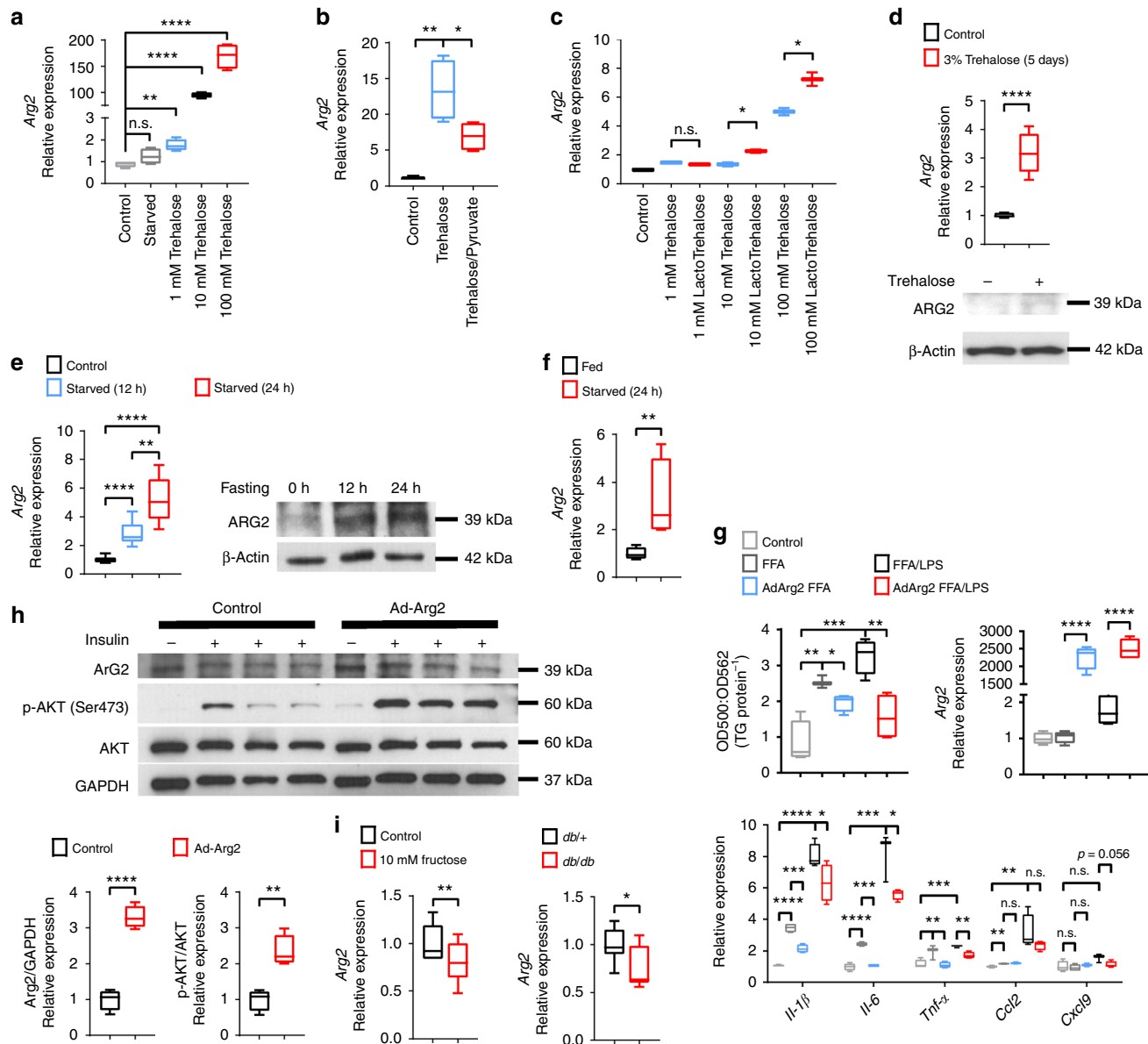

**Fig. 1** Arg2 is upregulated during fasting and trehalose treatment. **a** *Arg2* gene expression by qPCR analysis of cultured primary hepatocytes treated with 1–100 mM trehalose ($n = 6$ cultures per group, derived from three mice). **b** *Arg2* gene expression in response to 100 mM trehalose in the presence or absence of energetic reconstitution with pyruvate in primary hepatocytes ($n = 6$ hepatocyte cultures per group, derived from three mice). **c** *Arg2* gene expression by qPCR analysis of cultured primary hepatocytes treated with trehalose, or the trehalase-resistant analog, lactotrehalose ($n = 6$ cultures per group, derived from three mice). **d** Hepatic Arg2 expression by qPCR (top) and immunoblot (bottom) analysis of crude liver lysates from mice subjected to trehalose feeding (3% water ad libitum, 5 days) ($n = 5$ mice per group). **e** Arg2 expression by qPCR and immunoblot analysis of crude liver lysates from mice subjected to fasting ($n = 5$ mice per group). **f** Arg2 expression by qPCR analysis of primary hepatocytes subjected to low glucose and low serum treatment (24 h). **g** Left panels: in vitro enzymatic-colorimetric triglyceride accumulation assay and Right: *Arg2* gene expression in response to BSA-conjugated fatty acids with or without 1 nM lipopolysaccharide (LPS) and with or without adenoviral Arg2 overexpression in primary hepatocytes. Below: Inflammatory gene marker expression in parallel hepatocyte cultures analyzed above ($n = 6$ cultures per group, derived from three mice). **h** Immunoblot analysis of AKT phosphorylation in AML12 hepatocytes treated with or without 10 nM insulin following overnight serum deprivation ($n = 6$ cultures per group). **i** Arg2 gene expression in primary hepatocyte cultures following 24 h treatment with 10 mM fructose, or in crude liver lysates from *db/+* mice or *db/db* diabetic mice ($n = 4$ mice per group). For box plots, the midline represents the median, boxes represent the interquartile range and whiskers show the full range of values. *$P < 0.05$, **$P < 0.01$, ***$P < 0.001$, ****$P < 0.0001$ relative to bracketed control, by two-tailed Student's *t*-test

insulin-induced AKT phosphorylation (Fig. 1h, and Supplementary Fig. 6). Together, these data suggested that Arg2 over-expression cell autonomously mitigates hepatocyte fat accumulation, LPS-stimulated inflammatory responses and insulin sensitivity through the AKT pathway.

To define whether suppression of Arg2 correlates with metabolic disease, we compared hepatocyte Arg2 expression in fructose-treated and untreated primary hepatocyte cultures, and in LepR heterozygous (*db/+*) mice with LepR-deficient (*db/db*) diabetic mice, and in mice fed with a high-fat, high-sucrose diet (HFHS, 14 weeks). This revealed significantly reduced Arg2 expression in *db/db* mice and in response to fructose treatment (Fig. 1i). Therefore, Arg2 is induced during fasting and is reduced in states of nutritional excess.

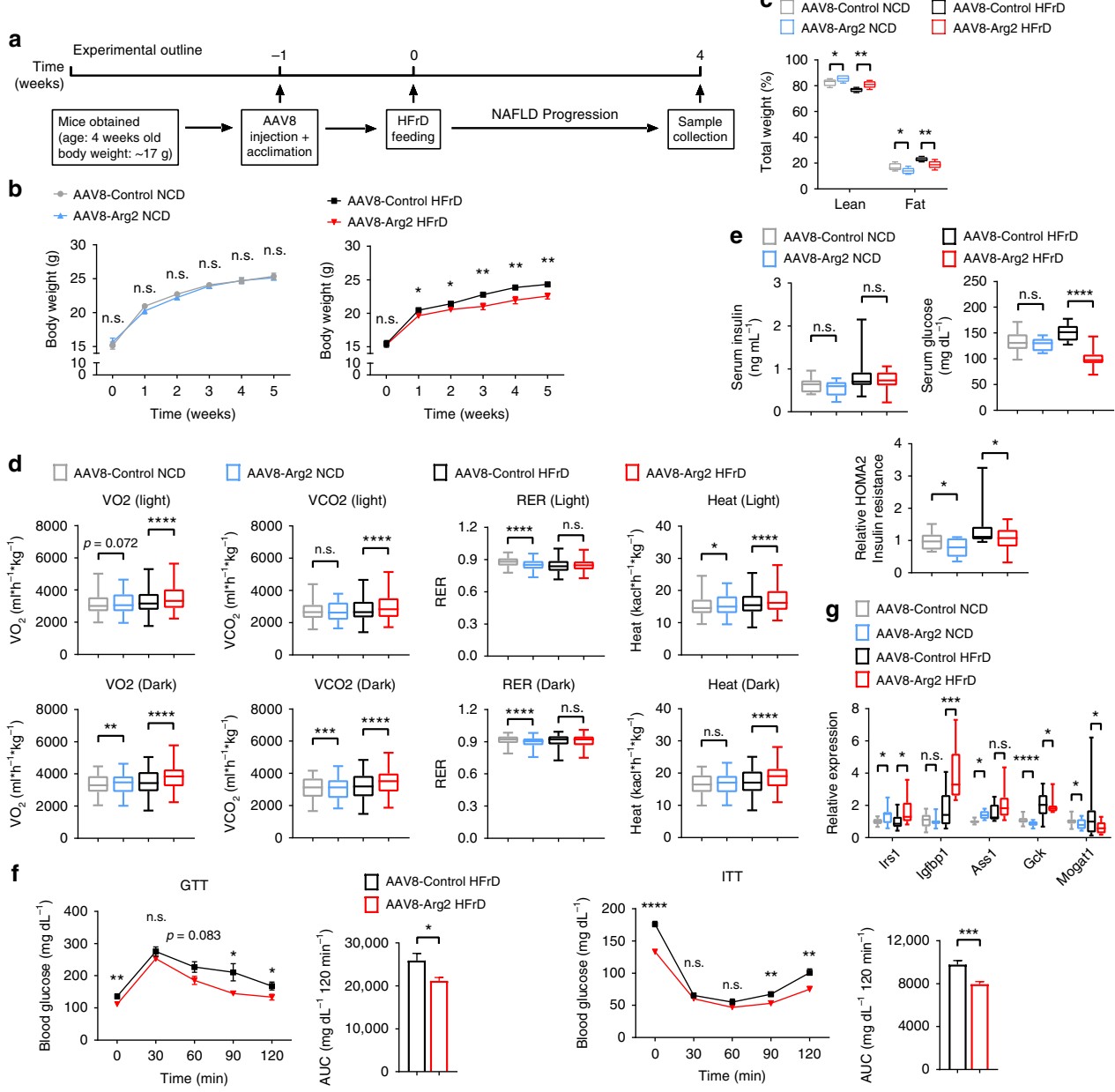

**Fig. 2** Hepatic Arg2 improves whole-body metabolism and insulin sensitivity. **a** Experimental design to test the role of Arg2 in protection from HFrD feeding. **b**, **c** Body weight (**b**) and echoMRI analysis of body composition (**c**) of AAV8-Control or AAV8-Arg2 WT mice fed NCD ($n = 6$–7 mice per group) or HFrD ($n = 8$ mice per group). **d** Indirect calorimetric quantification of $VO_2$, $VCO_2$, RER, and energy expenditure during light and dark cycles in AAV8-control and -Arg2 mice fed the indicated diet ($n = 6$ mice for NCD groups and $n = 8$ mice for HFrD groups). **e** Serum insulin (left), serum glucose (middle), and HOMA2 IR (right) in AAV8-control and -Arg2 mice fed NCD or HFrD. **f** Intraperitoneal glucose tolerance test (GTT) and insulin tolerance test (ITT) ($n = 5$ mice per group). **g** Hepatic mRNA expression of insulin-responsive genes. For box plots, the midline represents the median, boxes represent the interquartile range and whiskers show the full range of values. For bar graphs, data represent mean+s.e.m. *$P < 0.05$, **$P < 0.01$, ***$P < 0.005$, ****$P < 0.0001$ relative to vehicle treatment, by two-tailed Student's $t$-test

**Liver Arg2 blocks hepatic steatosis and insulin resistance**. Forced Arg2 expression in vitro reduced steatosis and enhanced insulin signaling. In vivo, hepatocyte-specific Arg2 over-expression was achieved by delivering adeno-associated virus 8 by tail-vein injection in mice (hereafter, AAV8-Arg2 mice). We fed these mice either a normal chow diet (NCD) or a 5-week high-fructose diet (HFrD, experimental design in Fig. 2a) to test whether forced Arg2 expression in vivo also enhances insulin signaling and reduces hepatic steatosis. We first demonstrated restricted hepatic overexpression of the Arg2 construct by qPCR and western blot analysis, without upregulation in skeletal muscle

or brown adipose tissue (Supplementary Fig. 1a). Importantly, we did not observe changes in hepatic *Arg1* mRNA expression in hepatic Arg2-overexpressing mice (Supplementary Fig. 1a). Hepatic Arg2 overexpression significantly increased lean mass and reduced fat mass in NCD and HFrD-fed mice (Fig. 2b, c). Moreover, Arg2 attenuated HFrD-induced weight gain without altering food intake (Supplementary Fig. 1b). Accordingly, indirect calorimetry revealed enhanced basal thermogenesis and oxygen and carbon dioxide exchange in AAV8-Arg2 versus HFrD-fed AAV8-controls (Fig. 2d), without altering light- or dark-cycle movement (Supplementary Fig. 1c).

The generalized fasting response induces peripheral lipolysis to generate substrate for liver fatty acid oxidation and ketogenesis. Analogously, hepatic Arg2 overexpression increased serum-free fatty acid (FFA) in AAV8-Arg2 HFrD mice with no significant changes in serum triglyceride (TG), total cholesterol, and LDL-C (Supplementary Fig. 1d). AAV8-Arg2 mice exhibited lower serum insulin and glucose concentrations in both NCD-fed and HFrD-fed group which resulted in lower HOMA of insulin resistance (HOMA-IR) (Fig. 2e). Glucose tolerance test (GTT) and insulin tolerance testing (ITT) confirmed improved glucose tolerance and insulin sensitivity secondary to hepatic Arg2 overexpression in HFrD-fed mice (Fig. 2f), without effects on glucose or insulin tolerance in NCD-fed mice (Supplementary Fig. 1e). With regard to insulin-responsive gene activation, Arg2 significantly increased expression of insulin receptor substrate 1 (*Irs1*), insulin-like growth factor binding protein 1 (*Igfbp1*), and argininosuccinate synthase 1 (*Ass1*). With regard to genes involved in de novo lipogenesis, only monoacylglycerol O-acyltransferase 1 (*Mogat1*) (Fig. 2g) was reduced in AAV8-Arg2 mice.

Hepatic analysis revealed lower liver weight and liver weight-to-body-weight ratios in HFrD-fed AAV8-Arg2 mice when compared with HFrD-fed AAV8-Controls, whereas no changes were detected between NCD-fed AAV8-Arg2 and AAV8-control groups (Supplementary Fig. 1f). Histological analysis of livers from control and AAV8-Arg2 mice by H&E and Oil Red O staining demonstrated that Arg2 attenuated HFrD-induced hepatic lipid accumulation (Fig. 3a). These findings were

confirmed by significantly reduced quantitative intrahepatic triglycerides, cholesterol and low-density lipoprotein cholesterol without changes in hepatic free fatty acids (FFA) (Fig. 3b). Although we observed no changes in serum ALT, AST or albumin in any group (Supplementary Fig. 1g), hepatic inflammatory gene expression (e.g., *Il-1β*, *Il-6*, *Tnfα*, *Ccl2*, *Cxcl9*) was significantly lower in the livers of AAV8-Arg2 mice when compared with AAV8-Controls in response to HFrD (Fig. 3c). Analysis of major metabolic pathways yielded no consistent changes in de novo lipogenic genes, import export (Cd36/FAT, Mttp), or fatty acid oxidation (*Acox1*, *Cpt1α*, *Cpt1β*, *Ucp2*, *Pparα*, *Pgc1α*, *Fgf21*) in HFrD-fed AAV8-Arg2 mice relative to littermate AAV8-Controls (Supplementary Fig. 1h–j).

**Hepatic Arg2 enhances thermogenesis in *db/db* diabetic mice.** To address the generalizability of the protective effects of hepatic Arg2, we explored the effects of Arg2 expression in a genetic model of obesity, leptin receptor-deficient (*db/db*) diabetic mice (Experimental outline in Fig. 4a). qPCR and immunoblot analysis confirmed restricted Arg2 overexpression in liver without Arg2 overexpression in visceral white adipose, brown adipose or skeletal muscle tissue (Supplementary Fig. 2a, b). Hepatic *Arg1* mRNA was increased by 10% uniquely in this genetically obese model (but not in the HFrD model), whereas hepatic Arg2 increased greater than 20-fold in *db/db* AAV8-Arg2 mice (Supplementary Fig. 2a). Hepatic Arg2 overexpression attenuated

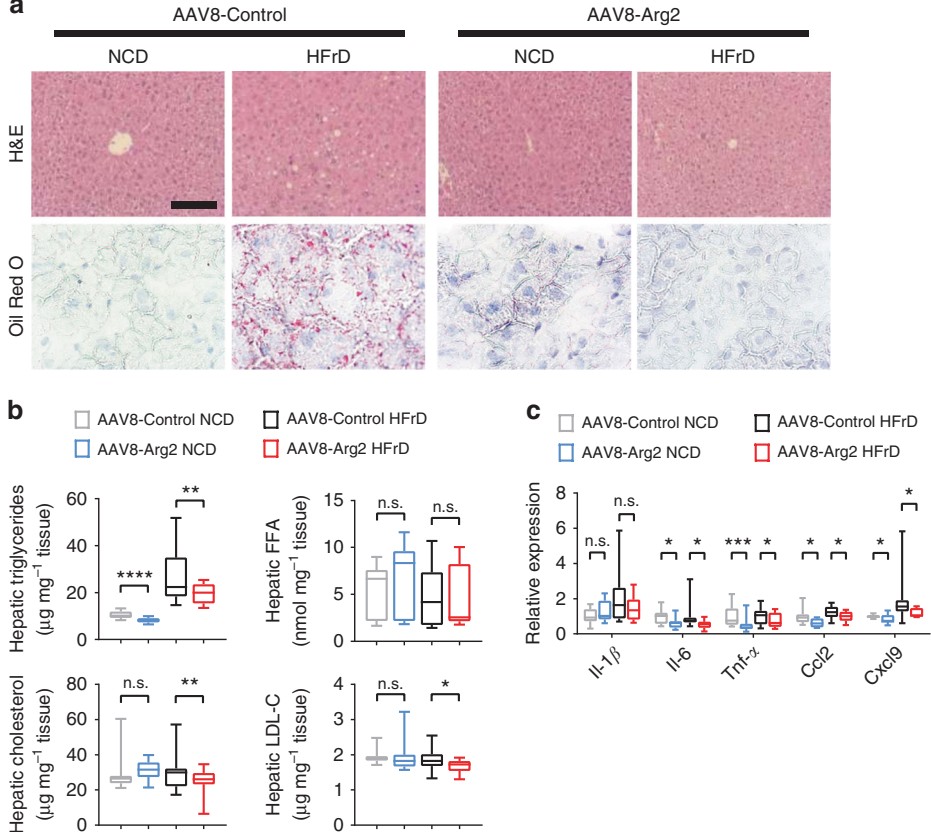

**Fig. 3** Hepatic Arg2 protects against hepatic steatosis and inflammation. **a** Representative hematoxylin and eosin (H&E)-stained (top) and Oil Red O-stained (bottom) liver sections from AAV8-Control (left) and AAV8-Arg2 (right) mice that were fed NCD or HFrD for 5 weeks. Scale bar, 100 μm. **b** Triglyceride, cholesterol, non-esterified fatty acid, and LDL-C contents in liver of AAV8-Control or AAV8-Arg2 injected WT mice that fed NCD (n = 6 to 7 mice per group) or HFrD (n = 8 mice per group). **c** Hepatic expression of genes involved in cytokine and chemokine inflammatory responses. Gene expression was normalized to 36B4 mRNA levels. For box plots, the midline represents the median, boxes represent the interquartile range, and whiskers show the full range of values. *P < 0.05, **P < 0.01, ***P < 0.005, ****P < 0.0001 relative to vehicle treatment, by two-tailed Student's t-test

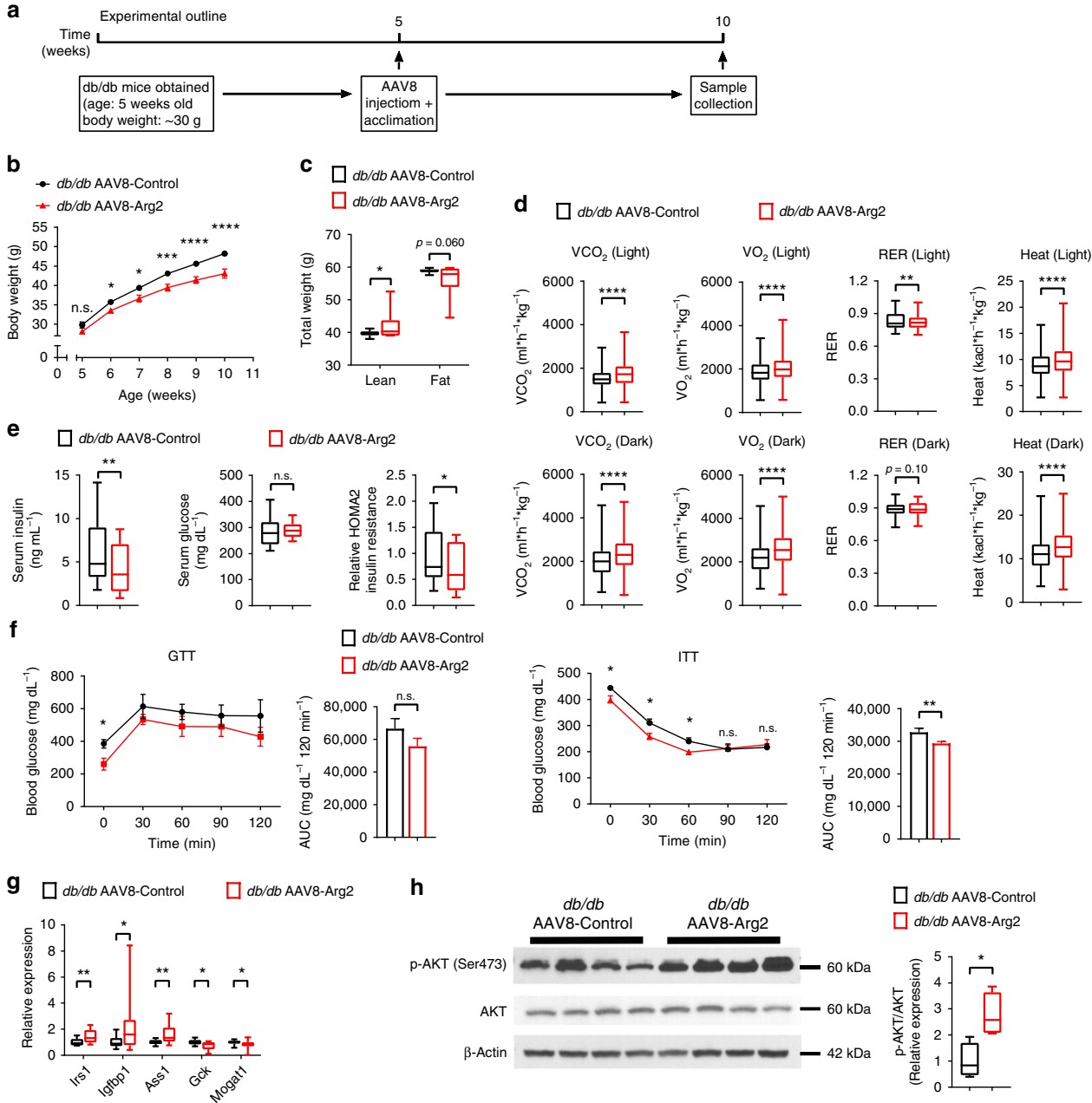

**Fig. 4** Hepatic Arg2 increases thermogenesis and insulin sensitivity in *db/db* mice. **a** Experiment schematic used to test the role of AAV8-mediated mouse Arg2 overexpression in *db/db* mice. **b**, **c** Body weight (**b**) and body composition (**c**) of AAV8-Control or AAV8-Arg2 injected *db/db* mice ($n = 8$ mice per group). **d** VO$_2$, VCO$_2$, energy expenditure, and RER during light and dark cycles in AAV8-injected *db/db* mice ($n = 8$ mice per group). **e** Serum insulin (left), glucose (middle), and HOMA2 IR (right) ($n = 8$ per group). **f** Intraperitoneal glucose tolerance test (GTT) (above) and insulin tolerance test (ITT) (below) ($n = 5$ per group). **g** Hepatic mRNA expression of insulin-responsive genes in liver samples from *db/db* AAV8-Control or AAV8-Arg2 mice ($n = 8$ mice per group). Gene expression was normalized to 36B4 mRNA levels. **h** Western blot analysis of total and phosphorylated AKT in liver samples from *db/db* AAV8-Control and AAV8-Arg2 mice ($n = 4$ mice per group). β-actin was probed as a loading control. For box plots, the midline represents the median, boxes represent the interquartile range and whiskers show the full range of values. For bar graphs, data represent mean+s.e.m. *$P < 0.05$, **$P < 0.01$, ***$P < 0.005$, ****$P < 0.0001$; n.s., not significant; relative to control treatment, by two-tailed Student's *t*-test

body weight gain in *db/db* AAV8-Arg2 mice compared with the *db/db* AAV8-Control mice (Fig. 4b) without changes in total food intake or locomotion (Supplementary Fig. 2c, d). Accordingly, *db/db* Arg2 mice had greater lean mass and lower fat mass (Fig. 4c), which corresponded with significantly higher VO$_2$, VCO$_2$, and energy expenditure in *db/db* AAV8-Arg2 mice versus *db/db* AAV8-Arg2 control mice. We did not observe changes in food consumption or movement (Fig. 4d, and Supplementary Fig. 2c, d). Hepatic Arg2 overexpression also increased circulating

TG, decreased total cholesterol and LDL-C, but it did not change FFA (Supplementary Fig. 2e). Analysis of insulin and glucose homeostasis revealed that serum insulin and homeostatic model assessment of insulin resistance (HOMA2-IR) were significantly decreased in *db/db* AAV8-Arg2 mice (Fig. 4e). Similarly, AAV8-Arg2-treated *db/db* mice exhibited significantly improved insulin tolerance testing (ITT) and trends toward improved glucose tolerance when compared with AAV8-Control injected *db/db* littermates (Fig. 4f). Consistent with these physiological findings,

gene expression of insulin-responsive genes, *Irs1*, *Igfbp1*, and *Ass1*, were elevated in livers of *db/db* AAV8-Arg2 mice versus AA8-Controls (Fig. 4g). *db/db* AAV8-Arg2 mice exhibited significantly enhanced hepatic AKT phosphorylation relative to *db/db* AAV8-Control littermates (Fig. 4h, and Supplementary Fig. 7). Similarly, analysis of iBAT, skeletal muscle and visceral adipose tissues each revealed trends toward increased AKT phosphorylation in *db/db* AAV8-Arg2 mice relative to *db/db* AAV8-Control littermates (Supplementary Fig. 2f).

**Hepatic Arg2 mitigates hepatic steatosis in *db/db* mice.** We examined the role of *Arg2* in mediating the major hallmarks of hepatic steatosis and inflammation. Similar to the observations made in HFrD-fed AAV8-Arg2 mice, hepatic *Arg2* overexpression decreased hepatic TG, FFA, and LDL-C content, without significantly altering total hepatic cholesterol in *db/db*

AAV8-Arg2 mice (Fig. 5a). Histological analysis of H&E and Oil Red O staining confirmed reduced lipid accumulation in *db/db* AAV8-Arg2 mice when compared with AAV8-control littermates (Fig. 5b). Biochemically, we observed lower serum ALT and trends toward lower AST with elevated serum albumin concentration in *db/db* AAV8-Arg2 mice relative to those *db/db* AAV8-Control mice (Fig. 5c). Attenuated steatosis in *db/db* AAV8-Arg2 mice was paralleled by decreased fatty acid synthetic genes (Fig. 5d), and fatty acid uptake and export genes (Fig. 5e). However, no consistent changes in gluconeogenic or fatty acid oxidation gene programs were observed (Supplementary Fig. 2g). Arg2 overexpression also attenuated the hepatic inflammatory gene expression response, as assessed by qPCR analysis of *Il-1β*, *Il-6*, *Tnf-α*, *Ccl2*, and *Cxcl9* in *db/db* mice as compared with their littermate controls at 10 weeks of age (Fig. 5f). Moreover, in hepatic tissue extracts and in serum, only hepatic alanine was significantly altered in *db/db* AAV8-Arg2 mice versus *db/db*

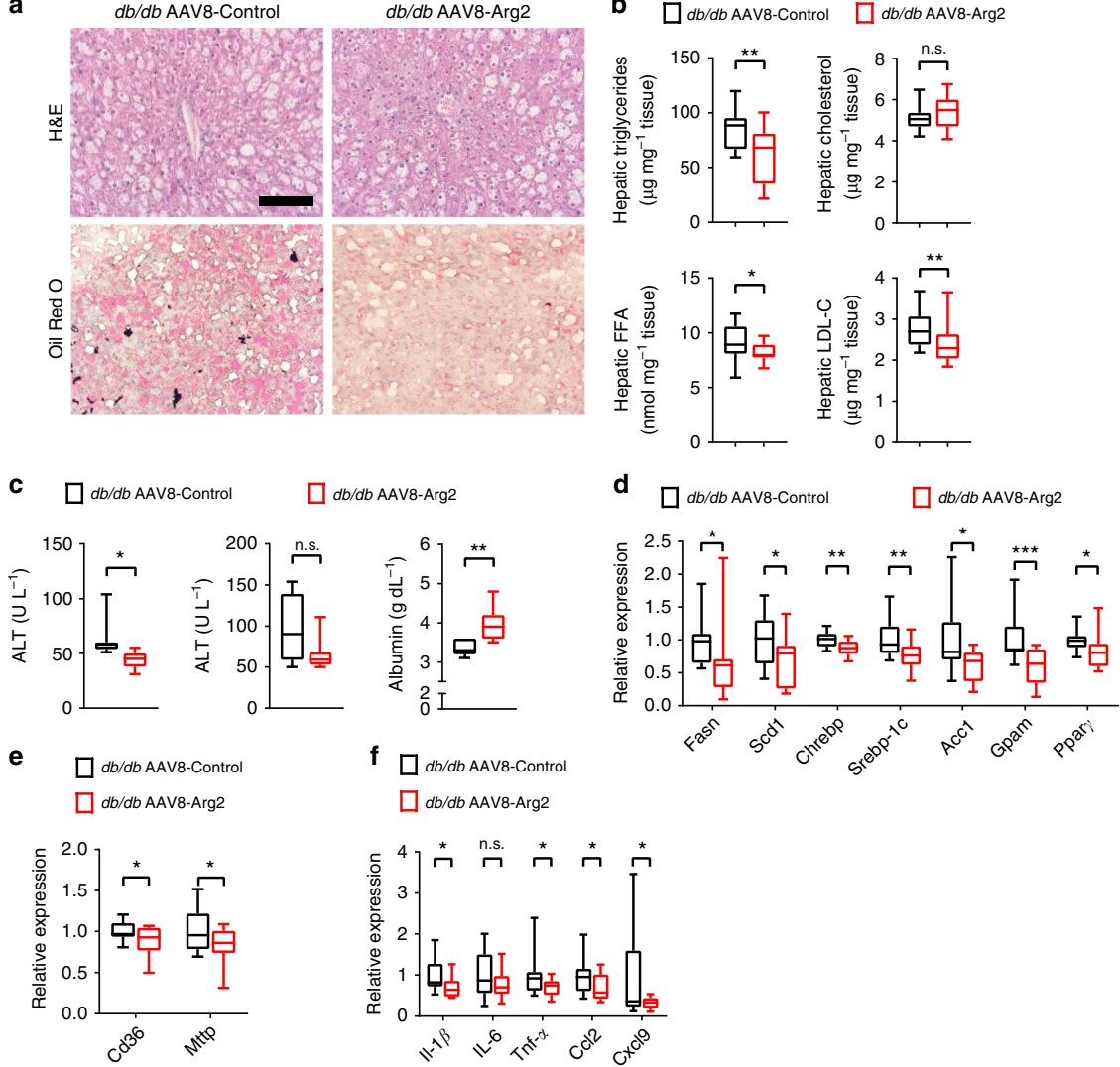

**Fig. 5** Hepatic Arg2 attenuates hepatic steatosis and inflammation in *db/db* mice. **a** Representative hematoxylin and eosin (H&E)-stained (top) and Oil Red O-stained (bottom) liver sections from AAV8-Control (left) and AAV8-Arg2 (right) *db/db* mice at 10 weeks of age. Scale bar, 100 μm. **b** Triglyceride, cholesterol, non-esterified fatty acid, and LDL-C contents in liver of AAV8-Control and AAV8-Arg2 injected *db/db* mice ($n = 8$ mice per group). **c** Serum ALT (left), AST (middle), and albumin (right) in *db/db* AAV8-Control and AAV8-Arg2 mice ($n = 7$ mice per group). **d, e** Hepatic mRNA expression of the indicated genes involved in de novo lipogenesis (**d**), and fatty acid uptake and export (**e**) in *db/db* AAV8-Control and AAV8-Arg2 mice ($n = 8$ mice per group). Gene expression was normalized to 36B4 mRNA levels. **f** Inflammatory mediator mRNA expression by qPCR in liver from *db/db* AAV8-Control and AAV8-Arg2 mice ($n = 8$ mice per group). For box plots, the midline represents the median, boxes represent the interquartile range and whiskers show the full range of values. *$P < 0.05$, **$P < 0.01$, ***$P < 0.005$, ****$P < 0.0001$; n.s., not significant; relative to control treatment, by two-tailed Student's *t*-test

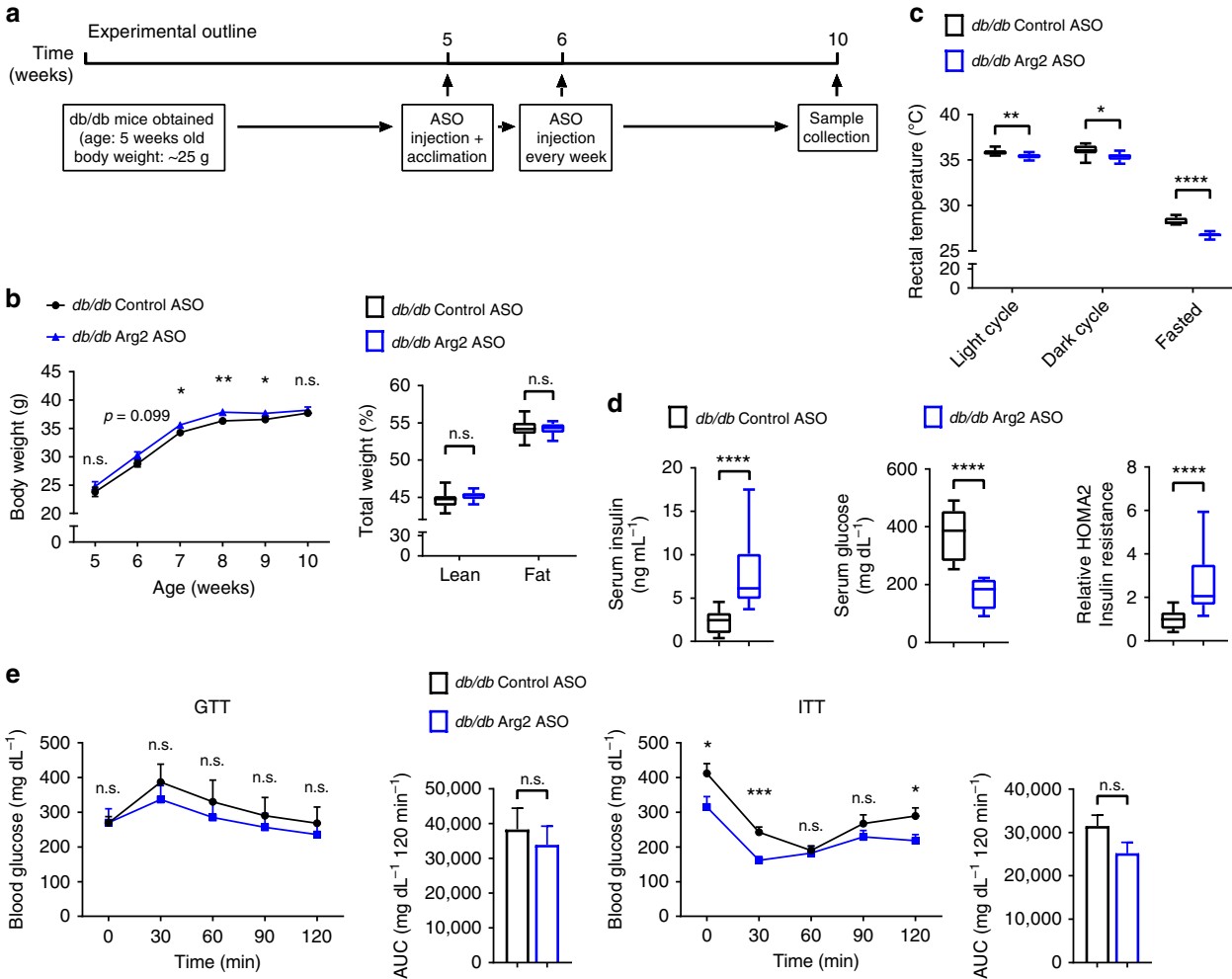

**Fig. 6** Hepatic Arg2 deficiency exacerbates hyperinsulinemia in *db/db* mice. **a** Experiment schematic used to test the role of hepatic Arg2 knockdown in *db/db* mice. **b** Body weight and body composition of Control ASO or Arg2 ASO-treated *db/db* mice (*n* = 8 mice per group). **c** Rectal temperature in *db/db* Control ASO or Arg2 ASO mice (*n* = 8 mice per group). **d** Serum insulin (left), glucose (middle), and HOMA2 IR (right) in Control ASO or Arg2 ASO-treated *db/db* mice (*n* = 8 mice per group). **e** Intraperitoneal glucose tolerance test (GTT) and insulin tolerance test (ITT) (*n* = 5 mice per group). For box plots, the midline represents the median, boxes represent the interquartile range and whiskers show the full range of values. For bar graphs, data represent mean+s.e.m. *$P < 0.05$, **$P < 0.01$, ***$P < 0.005$, ****$P < 0.0001$; n.s., not significant; relative to control treatment, by two-tailed Student's *t*-test

AAV8-Control mice after multiple comparison correction, as quantified by targeted metabolomic analysis (Supplementary Table 1). Surprisingly, no significant steady-state changes in urea cycle intermediaries or in arginine itself were defined in serum or hepatic extracts from *db/db* AAV8-Control and *db/db* AAV8-Arg2 mice.

**Higher cholesterol and insulin in Arg2ASO *db/db* mice.**
Hepatic Arg2 overexpression protected mice from diet- and genetically-induced insulin resistance, adiposity and hepatic steatosis and inflammation. To examine the consequences of liver-selective Arg2 deficiency, we treated *db/db* mice with either scrambled or Arg2-directed antisense oligonucleotides, followed by metabolic assessment (Fig. 6a). This resulted in significant hepatic Arg2 mRNA knockdown without knockdown in iBAT, SKM or WAT (Supplementary Fig. 3a) or in kidney (not shown). There were no hepatic changes in *Arg1* mRNA expression (Supplementary Fig. 3a). Endpoint total body weight, total fat mass, and fat percentage did not significantly change (Fig. 6b), although food consumption was slightly lower in liver Arg2-deficient mice (Supplementary Fig. 3b), and core temperature was

lower in *db/db* Arg2 ASO-treated mice when compared with *db/db* Control ASO-treated mice (Fig. 6c).

Quantifying serum parameters in *db/db* Arg2 ASO-treated mice revealed lower serum triglycerides and glucose, but elevated total cholesterol and LDL cholesterol as well as elevated insulin and calculated HOMA2 in *db/db* Arg2 ASO-treated mice when compared with *db/db* Control ASO-treated mice (Fig. 6d, and Supplementary Fig. 3c), although fasting glucose was lower in Arg2 ASO mice, in concordance with lower gluconeogenic gene (*G6pc* and *Fbp1*) expression (Fig. 6d, and Supplementary Fig. 3d). Total areas under the insulin- and glucose tolerance testing curves were unchanged in *db/db* Control versus Arg2 ASO-treated mice (Fig. 6e).

**Liver steatosis and inflammation after liver Arg2 knockdown.**
We next examined hepatic effects of liver Arg2 deficiency in *db/db* mice. Total liver weight and liver weight-to-body weight ratios were elevated in liver Arg2-deficient mice (Fig. 7a). Concomitantly, liver Arg2-deficient *db/db* mice exhibited increased neutral lipid accumulation, as determined by routine histological and Oil Red O analyses (Fig. 7b). Accordingly, hepatic

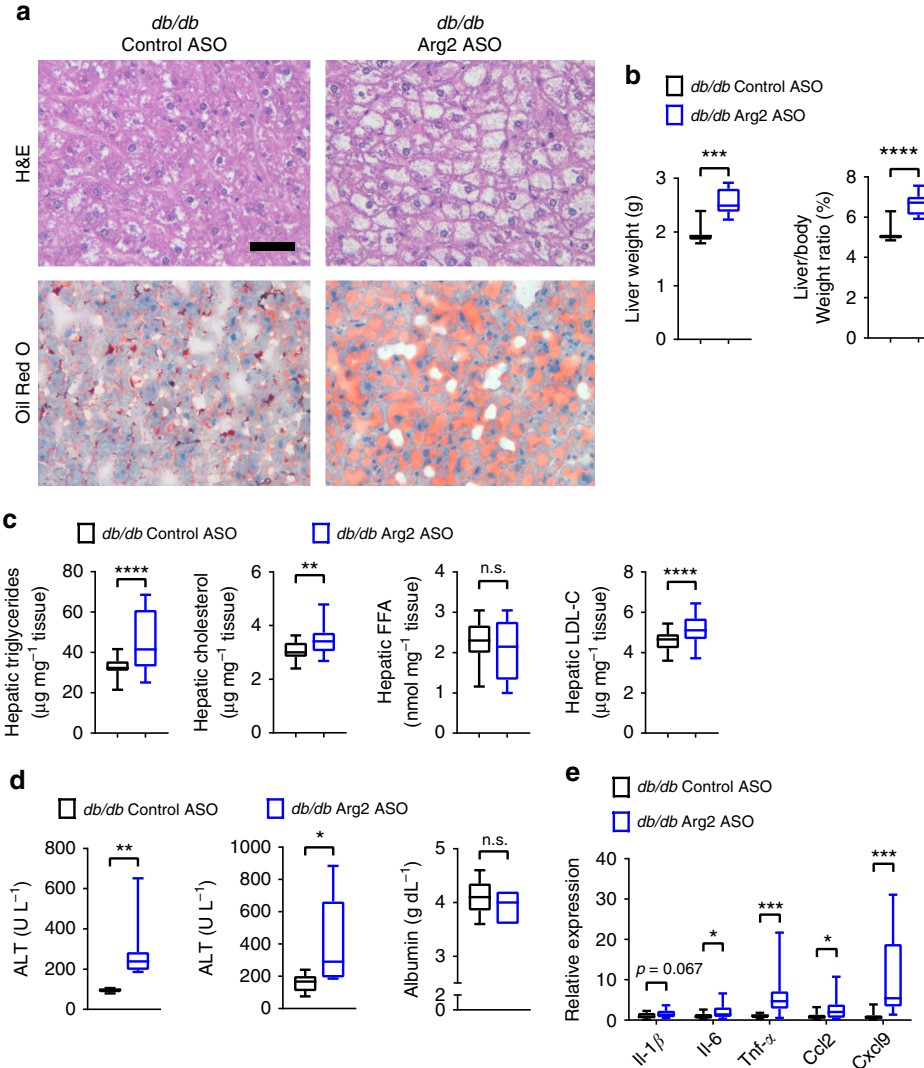

**Fig. 7** Hepatic Arg2 deficiency exacerbates hepatic steatosis and inflammation in *db/db* mice. **a** Representative hematoxylin and eosin (H&E)-stained (top) and Oil Red O-stained (bottom) liver sections from Control ASO and Arg2 ASO *db/db* mice at 10 weeks of age. Scale bar, 100 μm. **b** Liver weight and liver weight-to-body weight ratio in Control ASO and Arg2 ASO *db/db* mice. **c** Triglyceride, cholesterol, non-esterified fatty acid, and LDL-C contents in liver of AAV8-Control and AAV8-Arg2 injected *db/db* mice (*n* = 8 mice per group). **d** Serum ALT (left), AST (middle), and albumin (right) in *db/db* AAV8-Control and AAV8-Arg2 mice (*n* = 7 mice per group). **e** Hepatic mRNA expression of the indicated genes involved in hepatic inflammation in *db/db* Control and Arg2 ASO mice (*n* = 8 mice per group). For box plots, the midline represents the median, boxes represent the interquartile range and whiskers show the full range of values. *$P < 0.05$, **$P < 0.01$, ***$P < 0.005$, ****$P < 0.0001$; n.s., not significant; relative to control treatment, by two-tailed Student's *t*-test

triglycerides, total cholesterol and LDL-C and serum ALT and AST were all increased (Fig. 7c), without alterations in hepatic synthetic function in liver Arg2-deficient mice versus *db/db* controls (Fig. 7d). Hepatic inflammatory marker gene expression of *Il-6*, *Tnf-α*, *Ccl2*, and *Cxcl9* was increased in hepatic Arg2-deficient *db/db* mice versus *db/db* controls (Fig. 7e). Whereas gluconeogenic gene expression in Arg2-deficient *db/db* livers (Supplementary Fig. 3d), genes involved in de novo lipogenesis: *Scd1*, *Gpam*, and *Pparγ* were elevated in Arg2-deficient *db/db* livers when compared with *db/db* controls.

**Transcriptomics reveal inverse Arg2-Rgs16 regulation**. To explore potential mechanisms by which hepatic Arg2 over-expression improves hepatic lipid homeostasis, inflammation and insulin sensitivity, we examined unbiased transcriptional profiling by RNA-seq to analyze transcriptomic changes in the livers of *db/db* AAV8-Arg2 mice and AAV8-Control mice. Principal

component analyses revealed distinct gene clusters between *db/db* and *db/db* AAV8-Arg2 groups (Fig. 8a). This was corroborated by unsupervised gene-level clustering revealing distinctive patterns of significantly upregulated and downregulated genes when comparing *db/db* AAV8-Arg2 and *db/db* AAV8-Control mice (Fig. 8b). Kyoto Encyclopedia of Genes and Genomes (KEGG) pathway enrichment analysis of differentially expressed genes revealed enhanced phosphatidylinositol signaling, mTOR and FOXO signaling, consistent with enhanced hepatic insulin sig-naling. Conversely, consistent with fasting-state cellular physiol-ogy, significant downregulation of fructose, mannose, amino acid and nucleotide sugar metabolism, fatty acid elongation and bio-synthesis, and carbon/pyruvate metabolism (Fig. 8c). Volcano plot analysis confirmed upregulation of Arg2 gene expression in *db/db* AAV8-Arg2 mice, and identified the hepatocyte regulator of G-protein signaling 16 (RGS16) as among the most highly suppressed genes in *db/db* AAV8-Arg2 mice (Fig. 8d). In light of the described pathogenic role of RGS16 in hepatic steatosis[35], we

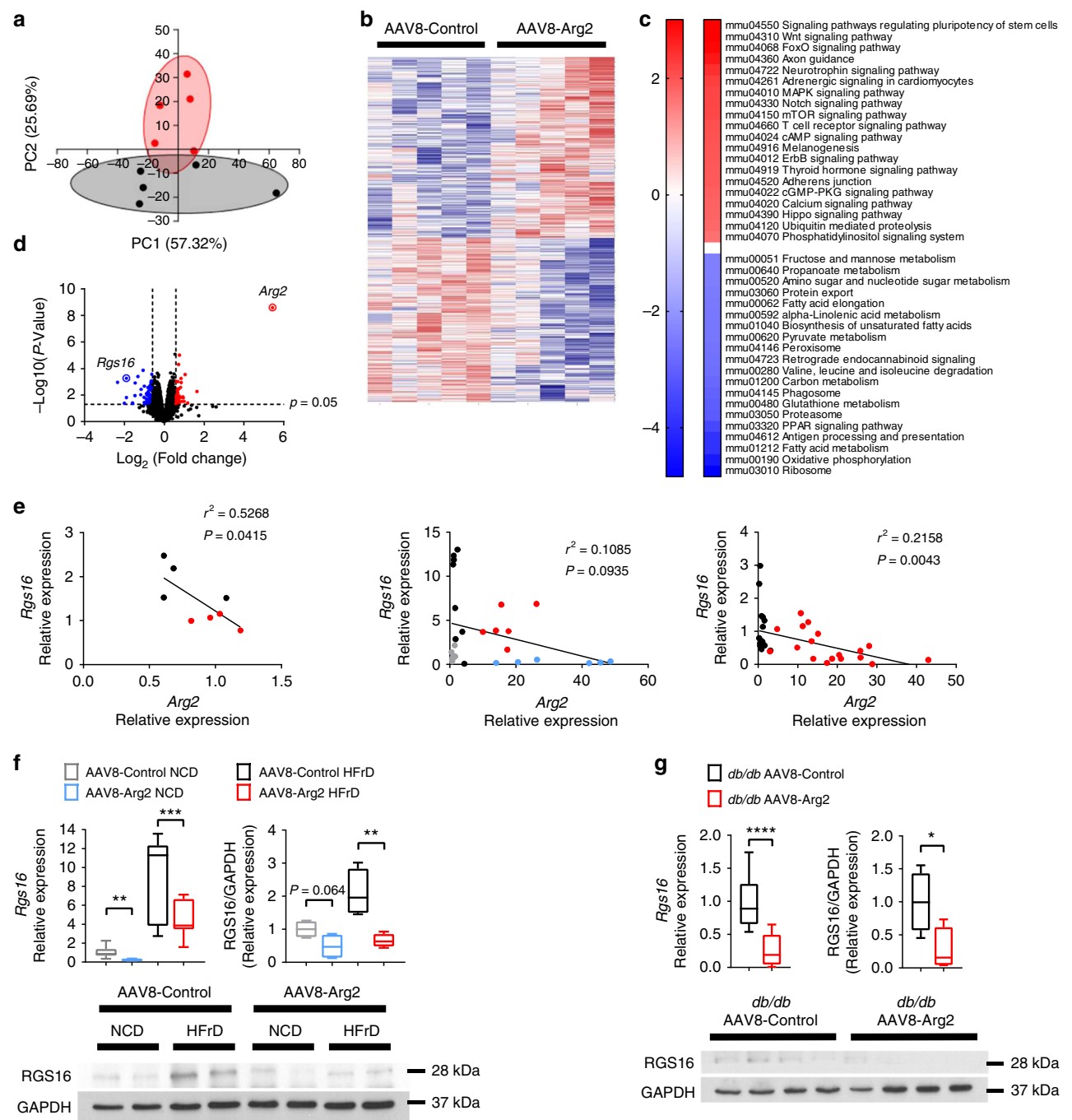

**Fig. 8** Inverse Arg2-RGS16 regulatory relationship is identified by unbiased transcriptomics. **a** Principal component analysis of transcriptomic data from *db/db* AAV8-Control and AAV8-Arg2 livers. **b** Heat map demonstrating gene-level regulation in *db/db* AAV8-Control and AAV8-Arg2 livers. **c** KEGG pathway analysis demonstrating top up- and downregulated gene pathways in *db/db* AAV8-Arg2 livers. **d** Volcano plot demonstrating significantly altered genes by expression level and *P*-value. RGS16 and Arg2 are highlighted as highly regulated genes identified within the dataset. **e** Linear regression analysis demonstrating inverse Arg2 and RGS16 correlation in (left to right:) *db/+* versus *db/db* livers, in HFrD-fed AAV8-control and AAV8-Arg2 mice, and in *db/db* AAV8-control and AAV8-Arg2 mice. **f, g** RGS16 gene expression and protein abundance in HFrD-fed (**f**) and *db/db* (**g**) AAV8-control and AAV8-Arg2 mouse livers. For box plots, the midline represents the median, boxes represent the interquartile range and whiskers show the full range of values. *$P < 0.05$, **$P < 0.01$, ***$P < 0.005$, ****$P < 0.0001$; n.s., not significant; relative to control treatment, by two-tailed Student's *t*-test

further interrogated whether Arg2 and RGS16 anti-correlate in multiple models of metabolic disease. In *db/+* mice, in wild-type mice overexpressing Arg2 and in *db/db* AAV8-Arg2 mice, we observed an inverse relationship between Arg2 and Rgs16 expression (Fig. 8e). Moreover, hepatic *Rgs16* mRNA and protein levels were significantly suppressed by direct Arg2 expression in the livers of chow and HFrD-fed and *db/db* AAV8-Arg2 mice (Fig. 8f, g, and Supplementary Fig. 8), whereas hepatic Arg2-

deficient mice exhibited upregulated hepatic *Rgs16* mRNA (Supplementary Fig. 4). Together, the data identify an inverse relationship between hepatic Rgs16 and Arg2.

**Hepatic Rgs16 reconstitution reverses hepatic Arg2 effects.** We next examined the hypothesis that genetic reconstitution of hepatic RGS16 in the context of Arg2 overexpression would

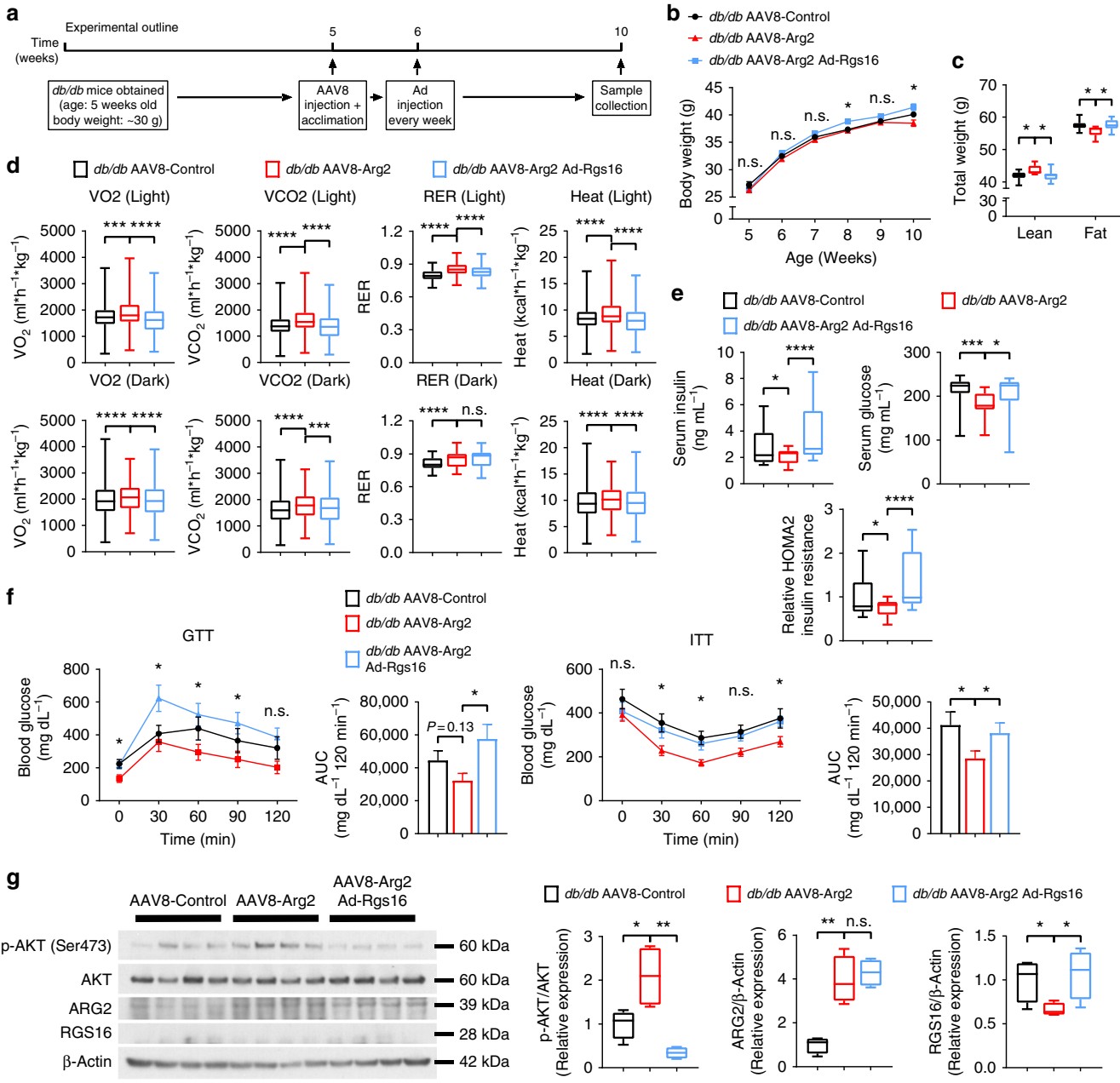

**Fig. 9** Hepatic RGS16 reconstitution reverses *Arg2*-mediated improvements in peripheral fat accumulation and glucose homeostasis. **a** Experimental schematic used to test the role of AAV8-mediated mouse Arg2 and Ad-RGS16 overexpression in *db/db* mice. **b** Body weight and **c** body composition of AAV8-Control or AAV8-Arg2 *db/db* mice expressing Ad-Control or Ad-RGS16 (n = 8 per group). **d** VO2, VCO2, energy expenditure, and RER during light and dark cycles in AAV8-Control or AAV8-Arg2 *db/db* mice expressing Ad-Control or Ad-RGS16 (n = 8 per group). **e** Serum insulin (left), glucose (middle), and HOMA2 IR (right) in AAV8-Control or AAV8-Arg2 *db/db* mice expressing Ad-Control or Ad-RGS16 (n = 8 per group). **f** Intraperitoneal GTT and ITT (n = 5 mice per group). **g** Western blot analysis of total and phosphorylated AKT in liver samples from *db/db* AAV8-Control and AAV8-Arg2 mice (n = 4 mice per group). For box plots, the midline represents the median, boxes represent the interquartile range and whiskers show the full range of values. For bar graphs, data represent mean+s.e.m. *$P < 0.05$, **$P < 0.01$, ***$P < 0.005$, ****$P < 0.0001$; n.s., not significant; relative to control treatment, by two-tailed Student's *t*-test

reverse the improvements in hepatic steatosis, inflammation and insulin and glucose intolerance observed in Arg2-treated *db/db* mice. To that end, we treated *db/db* AAV8-Control and AAV8-Arg2 mice with adenovirus encoding GFP (Ad-Control) or RGS16 (Ad-*Rgs16*, Experimental outline, Fig. 9a). We confirmed Arg2 overexpression, Rgs16 suppression and mild Rgs16 reconstitution in response to our AAV8 and adenoviral manipulation (Supplementary Fig. 5a, b). In *db/db* mice, Arg2 overexpression attenuated weight gain, increased fat mass and decreased lean mass, although we noted the weight bifurcation occurred later in the

adenovirus-treated cohorts (Fig. 9b) than in the control cohorts (Fig. 4b). Nevertheless, these changes were reversed by RGS16 reconstitution in *db/db* AAV8-Arg2 mice when compared with Ad-Control-treated mice (Fig. 9b, c). *db/db* AAV8-Arg2 mice again exhibited enhanced energy expenditure and oxygen consumption by indirect calorimetry when compared with *db/db* AAV8-control mice without altering locomotion or food consumption (Fig. 9d, and Supplementary Fig. 5c, d). However, RGS16 reconstitution reversed the energetic enhancements in *db/db* AAV8-Arg2 mice (Fig. 9d). Accordingly, Rgs16

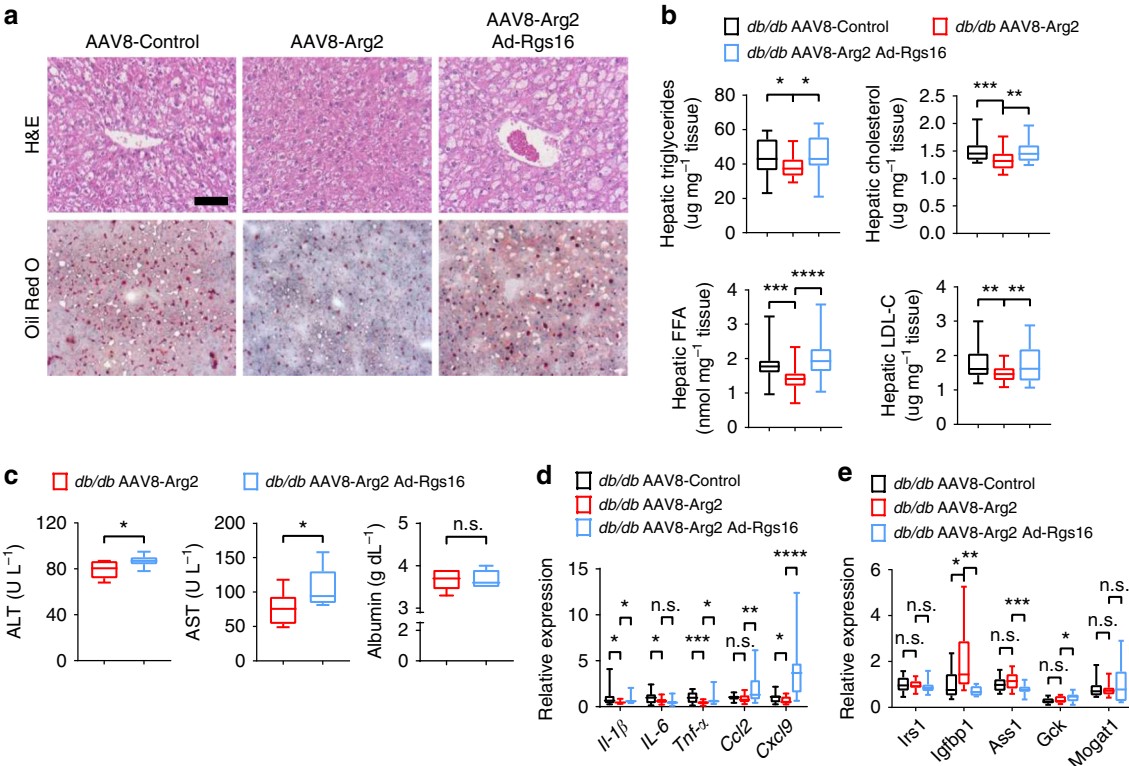

**Fig. 10** Hepatic RGS16 reconstitution reverses *Arg2*-mediated improvements in hepatic fat accumulation. **a** H&E and Oil Red O-stained liver sections from *db/db* AAV8-Control and AAV8-Arg2 mice treated with Ad-GFP or Ad-RGS16. Scale bar, 100 μm. **b** Hepatic triglycerides, cholesterol, non-esterified fatty acids and LDL-C in livers from *db/db* AAV8-Control and AAV8-Arg2 mice treated with Ad-GFP or Ad-RGS16. **c** Serum ALT, AST and albumin in *db/db* AAV8-Arg2 mice expressing Ad-GFP or Ad-RGS16. **d** Hepatic inflammatory gene marker mRNA by qPCR. **e** mRNA expression of insulin-responsive genes by qPCR ($n = 8$ mice per group). For box plots, the midline represents the median, boxes represent the interquartile range and whiskers show the full range of values. *$P < 0.05$, **$P < 0.01$, ***$P < 0.005$, ****$P < 0.0001$; n.s., not significant; relative to control treatment, by two-tailed Student's *t*-test

reconstitution substantially exacerbated insulin resistance in *db/db* mice as defined by serum insulin, glucose, and HOMA2-IR (Fig. 9e). This was further confirmed by glucose and insulin tolerance testing (Fig. 9f). Hepatic insulin sensitivity, as quantified by insulin-stimulated AKT phosphorylation, was impaired by Rgs16 expression in the liver (Fig. 9g, and Supplementary Fig. 9) when compared with *db/db* AAV8-Arg2 mice treated with Ad-Control.

In the liver, Arg2 decreased hepatic TG, cholesterol, FFA, and LDL-C in *db/db* Arg2 mice, and these effects were significantly reversed upon Rgs16 reconstitution in *db/db* AAV8-Arg2 mice (Fig. 10a) with no significant changes detected in serum TG, cholesterol, FFA and LDL-C (Supplementary Fig. 5e). Histological analysis by H&E and Oil Red O demonstrated that Arg2 overexpression reduced hepatic lipid accumulation, which was reversed in *db/db* mice also expressing AdRGS16 (Fig. 10b). Similarly, serum ALT, AST as well as hepatic inflammatory marker expression in *db/db* mice were each significantly elevated in *db/db* AAV8-Arg2 mice expressing Ad-Rgs16 as compared with *db/db* AAV8-Arg2 Ad-Control littermates (Fig. 10c, d). Rgs16 overexpression reversed Arg2 effects on *Igfbp1* expression, and RGS16 overexpression generally increased genes involved in de novo lipogenesis in liver, including *Chrebp, Acc1, Gpam,* and *Pparγ* (Fig. 10e). However, the expression of mRNAs encoding beta oxidation and fatty acid uptake and export were largely unchanged by hepatic Arg2 overexpression or Rgs16 reconstitution (Supplementary Fig. 5f).

### Discussion
Arginases are ancient, highly conserved enzymes[36,37], the purpose of which is long considered to be conversion of L-arginine to urea

and ornithine[36–43]. More recently, these enzymes have been implicated in the biosynthesis of polyamines, and in proline, glutamate, creatine, and agmatine biosynthesis as well[38,40,42,44]. The two vertebrate arginases, type I and type II evolved separately from an early gene duplication event[36], which ultimately manifests as both distinct and overlapping enzymatic functions, which yet remain to be fully appreciated[42]. We demonstrate here that Arg2 is more profoundly induced during fasting in liver versus Arg1. This opens the possibility now that the arginases regulate fasting biology in the liver. Although the total abundance of Arg1 is much greater than that of Arg2 in the basal setting[42], the data here suggest that Arg1 and Arg2 are in fact differentially regulated during fasting, and may thus serve divergent functions as part of the physiological fasting response. The fundamental biological purpose of hepatocyte Arg2 regulation during fed and fasting states, and during health and disease remains an intriguing area of further exploration.

The fasting response enhances hepatic and extrahepatic energy homeostasis, and yet the intermediaries that dictate these effects remain poorly understood. Here, we demonstrate that hepatocyte arginase 2 is upregulated during fasting, and after trehalose and trehalose analog treatment. Furthermore, Arg2 overexpression per se attenuates insulin resistance, hepatic steatosis and inflammation in high-fructose-fed and genetically obese models. Arg2 is therefore a tractable target that can be used to leverage the therapeutic hepatic fasting response. Although it must be noted that some of the individual metabolic improvements observed upon AAV8-Arg2 expression were modest. For example, we observed 20–25% improvements in glucose and insulin tolerance in HFrD-fed AAV8-Arg2 and *db/db* x AAV8-Arg2 mice.

However, it should also be noted that the magnitude of changes in glucose and insulin tolerance here mirrors that observed in humans subjected to short-term intermittent fasting or caloric restriction, which results in 11–30% improvements in HOMA-IR, and 11–40% improvement in fasting plasma insulin levels[7]. Indeed even high-intensity, supervised diet and exercise training regimens over a 4-month period produced even more modest improvements in glucose tolerance, and only for the highest-intensity intervention groups[45].

The observation that hepatic Arg2 activation is therapeutic originated from prior observations that hepatic fasting signals (e.g., TFEB, PPARα) mediated the therapeutic effects of genetic or pharmacological glucose transport blockade[27,29]. However, the observation that hepatic glucose uptake is impaired in diabetic patients[46] may raise an apparent paradox, in light of the intrinsic therapeutic effect of hepatic glucose transport blockade. At least two key differences between the fasted liver and the diabetic liver resolve this apparent paradox. First, insulin stimulates hepatic glucose uptake by twofold over the fasting state[47]. In diabetic patients, insulin-stimulated glucose uptake is impaired (by 17.4% in the Iozzo study[46]) but this degree of blunted fed-state glucose uptake would not be expected to mimic a true hepatocyte fasting response. Second, *basal* hepatic glucose uptake should remain elevated through hepatic facilitative transporters GLUT1, GLUT2, and GLUT8. This is because basal hyperglycemia in the diabetic patient provides a glucose concentration gradient that drives these facilitative transporters even in the absence of insulin-stimulated glucose transport. This rise in basal hepatic glucose uptake would be expected to further impede the hepatocyte fasting response. Thus, although insulin-stimulated glucose uptake is relatively impaired in the insulin-resistant state, our data suggest that pharmacological hepatic GLUT blockade or targeted hepatic Arg2 activation can drive the therapeutic hepatic glucose response in diet-induced and genetic models of obesity and insulin resistance.

Taken further in context, Arg2 appears to serve tissue-specific, and context-dependent functions in energy homeostasis. These context-dependent functions underscore the importance of leveraging hepatocyte fasting responses over less targeted approaches to metabolic therapy. Endothelial Arg2 over-expression surprisingly impaired endothelial cell autophagy and AMPK signaling[48], and macrophage Arg2 appears to have pro-inflammatory functions[49–51], whereas germline Arg2 deletion effects are not yet clear[49,52]. The clinical implications of these data should thus be underscored, especially in view of recent momentum to inhibit arginases in the setting of cardiovascular and other disease[53]. Our liver-deficient Arg2 model suggests that blocking Arg2 without sufficient specificity may have negative metabolic consequences, particularly if hepatic Arg2 is inadvertently targeted. Here, we utilized a targeted, hepatocyte-specific overexpression approach to demonstrate further adaptive metabolic effects of hepatic Arg2 overexpression. Mechanistically, our genetic complementation data indicate that Arg2 cell autonomously enhances hepatic insulin signaling in vitro and in vivo and reduces hepatic fat accumulation via Rgs16 suppression. The data give rise to the hypothesis that Rgs16 suppresses hepatocyte-protective G-protein-coupled receptor signaling events, the origin and nature of which remain to be addressed.

RGS16 was previously identified as the only one of 21 RGS genes that was regulated by circadian rhythms in the murine suprachiasmatic nucleus and liver[54,55]. Hepatic Rgs16 was subsequently determined to suppress hepatic fat oxidation by upregulating the carbohydrate response element binding protein[35]. Because the net physiological response to fasting in the liver is to upregulate fat oxidation, the proposed Rgs16 function is to exert

negative feedback on Gi/Gq-mediated fat oxidation[35,54]. This negative feedback appropriately moderates the prolonged fasting response (e.g., during hibernation). In overfed contexts, in which excess caloric and fat dissipation is an appropriate full-time compensation, RGS16 function becomes maladaptive. The maladaptive nature of this regulation is supported by comparing Arg2 expression in *db/db* vs *db/+* mice, HFrD-fed vs Chow-fed mice and in primary hepatocyte cultures treated with or without 10 mM fructose. In each of these overfed contexts, Arg2 expression was suppressed. Our data demonstrating that Arg2 acts upstream of Rgs16 to suppress its induction may reflect a molecular bypass in this negative feedback loop. Clinically, this Arg2-regulated bypass route may be exploited to treat NAFLD and insulin resistance.

In summary, this report identifies hepatic Arg2 suppression of Rgs16 as a metabolic regulatory axis during fasting. Moreover, we identify pharmacologic fasting mimetics—trehalose and lacto-trehalose—which activate this axis analogous to physiological fasting. Therefore, the hepatic intermediaries that mediate the generalized macronutrient fasting response can be directly leveraged to treat cardiometabolic disease.

## Methods

**Cell lines.** AML12 cells (CRL-2254) were purchased directly from the American Type Culture Collection (ATCC) and propagated and maintained precisely per manufacturer specification.

**Primary hepatocyte cell isolation, culture and treatment.** Primary murine hepatocytes obtained from WT mice were isolated[24,25,31] and cultured and maintained in regular DMEM growth media (Sigma, #D5796) containing 10% FBS. For in vitro starvation experiments, "starved" media contained 1 g/L glucose and 0.5% FBS was used. Cultures were lysed in Trizol and subjected to downstream analysis. In vitro genetic knockdown was achieved via siRNA transfection using Lipofectamine 3000 from Invitrogen (L3000015). Trehalose was obtained from Sigma Aldrich (St. Louis, MO) and was >97% purity by HPLC. Three percent trehalose water (w/v) fed ad libitum was used in all in vivo experiments.

**In vitro insulin signaling.** AML12 cells were seeded in 6-well plates. Cells were infected with adenoviruses, Ad-GFP, or Ad-Arg2, for 24 h in regular DMEM HAM/F12 growth media and starved in serum-free DMEM HAM/F12 media for 16 h before being treated with either serum-free media or serum-free media with insulin (10 nM) for 10 min.

**Animal studies.** All animal protocols were approved by the Washington University School of Medicine Animal Studies Committee. Male C57B/6J mice and *db/db* mice were purchased directly from the Jackson Laboratory (Bar Harbor, ME) and housed a 12-h alternating light-dark, temperature-controlled, specific pathogen-free barrier facility prior to and throughout experimentation.

All animals received humane care and procedures were performed in accordance with the approved guidelines by the Animal Studies Committee at Washington University School of Medicine. All animal studies were performed in accordance with the criteria and ethical regulations outlined by the Institutional Animal Care and Use Committee (IACUC).

**Quantitative real-time RT-PCR (qRT-PCR).** Total RNA was prepared by homogenizing snap-frozen livers or cultured hepatocytes in Trizol reagent (Invitrogen #15596026) according to the manufacturer's protocol. cDNA was prepared using Qiagen Quantitect reverse transcriptase kit (Qiagen #205310). Real-time qPCR was performed with Step-One Plus Real-Time PCR System (Applied Biosystems) using SYBR Green master Mix Reagent (Applied Biosystems) and specific primer pairs. Relative gene expression was calculated by a comparative method using values normalized to the expression of an internal control gene. Primers are shown in Supplementary Table 2.

**Immunoblotting.** Tissues were homogenized in RIPA buffer supplemented with protease and phosphatase inhibitors (Thermo Scientific). After homogenization, lysate was centrifuged at $18,000 \times g$ for 15 min at 4 °C, and the supernatant was recovered. Protein concentration was determined by BCA Assay Kit (Thermo Scientific) and was adjusted to 2 mg/mL. Samples for western blotting were prepared by adding Laemmli buffer at a ratio of 1:1 and heating at 95 °C for 5 min. The prepared samples were subjected to 10% or 13% SDS-PAGE, followed by electrical transfer onto a nitrocellulose membrane using the Trans-Blot Turbo system (Bio-Rad). After blocking the membrane with 5% milk in TBST, the

membrane was incubated in primary antibody at 4 °C overnight. The blot was developed after secondary antibody incubation using Pierce ECL Western Blotting Substrate (Thermo Scientific). Blots were developed according to the manufacturer's instructions. Protein expression levels were quantified with Image Lab software and normalized to the levels of β-actin or GAPDH.

**Antibodies.** Antibodies against phosphor-IRS1 (Ser636/639) (no. 2388), IRS1 (no. 2382), phosphor-AKT (Ser473) (no. 9271), AKT (no. 9272), GAPDH (no. 5174), and β-actin (no. 3700) were purchased from Cell Signaling Technology (CST) (Beverly, MA, USA). Anti-Arg2 antibody was purchased from Santa Cruz Biotechnology (no. Ab154422). Antibody against RGS16 (no. NBP2-01584) was purchased from Novus Biologicals, LLC (no. Ab203071) (Littleton, CO, USA). The dilution ratio for all primary antibodies was 1:1000. The secondary antibodies used in this study were peroxidase-conjugated anti-rabbit IgG (A7074) and anti-mouse IgG (A7076) purchased from CST, in which were used at a 1:5000 dilution.

**Histological analysis.** Formalin-fixed paraffin-embedded liver sections were stained by H&E via the Washington University Digestive Diseases Research Core Center. OCT-embedded frozen liver sections were stained by Oil Red O according to standard protocols flowered by microscopic examination. Three liver sections were examined and evaluated for each animal. For Oil red O staining, ice-cold methanol-fixed frozen sections from mice were stained according to described protocols [24,25,31].

Lactotrehalose synthesis and purification was carried out using chemoenzymatic synthesis and purification methods we described [56]. Reactions were performed in 50 mM HEPES buffer (pH 7.4) containing 10 mM galactose, 40 mM UDP-glucose, 20 mM MgCl$_2$, and 9.8 μM TreT at 70 °C. We confirmed 98–99% purity by $^1$H-NMR (not shown). We confirmed 98–99% purity by $^1$H-NMR prior to experiments.

**AAV8- and adenovirus-mediated overexpression.** Serotype 8 AAV (AAV8) was administered via tail vein as we previously reported [29]. All viral vectors (RGS16, Arg2, shTFEB) were obtained directly from Vector Biolabs Inc (Philadelphia, PA).

**Antisense oligonucleotides.** We obtained validated Arg2-specific antisense oligonucleotides from IONIS Pharmaceuticals (Carlsbad, CA). Mice were treated precisely as we previously reported [27,32], prior to in vivo and post-mortem assays.

**Insulin and glucose tolerance testing.** For insulin tolerance tests (ITT), mice were injected with 0.75 IU per kg body weight of insulin (Humalog, Eli Lilly) intraperitoneally after 4 h of fasting on Aspen bedding. For glucose tolerance tests (GTT), mice were injected with 2 g per kg body weight of glucose intraperitoneally after fasting for 4 h on aspen bedding. *db/db* mice were injected with 1 g per kg body weight of glucose intraperitoneally after fasting for 16 h on aspen bedding. Blood samples were measured at different time points with a glucometer (Arkray USA, Inc., Minneapolis, MN, USA).

**Clinical chemistry measurements and hepatic lipid analyses.** For all other serum analyses, submandibular blood collection was performed immediately prior to sacrifice and serum was separated. Insulin ELISA (Millipore #EZRMI-13K), triglycerides (Thermo Fisher Scientific #TR22421), cholesterol (Thermo Fisher Scientific #TR13421), and free fatty acids (Wako Diagnostics #999-34691, #995-34791, #991-34891, #993-35191) quantification were performed using commercially available reagents according to manufacturer's directions. Albumin levels were quantified using an AMS LIASYS Chemistry Analyzer.

Hepatic lipids were extracted from ~100 mg hepatic tissue homogenized in 2:1 chloroform:methanol. In total, 0.25–0.5% of each extract was evaporated overnight prior to biochemical quantification of triglycerides, LDL-C, cholesterol, and free fatty acids using reagents described above, precisely according to manufacturer's directions.

**Body composition analysis.** Body composition analysis was carried out in unanesthetized mice using an EchoMRI 3-1 device (Echo Medical Systems) via the Washington University Diabetic Mouse Models Phenotyping Core Facility.

**Indirect calorimetry and food intake measurement.** All measurements were performed in a PhenoMaster System (TSE systems) via the Washington University Diabetic Mouse Models Phenotyping Core Facility, which allowed metabolic performance measurement and activity monitoring by an infrared light = beam frame. Mice were placed at room temperature (22–24 °C) in separate chambers of the PhenoMaster open-circuit calorimetry. Mice were allowed to acclimatize in the chambers for 4 h. Food and water were provided ad libitum in the appropriate devices. The parameters of indirect calorimetry (VO$_2$, VCO$_2$, respiratory exchange ratio (RER), heat and movement) were measured for at least 24 h for a minimum of one light cycle (6:01 am to 6:00 pm) and one dark cycle (6:01 pm to 6:00 am). Presented data are average values obtained in these recordings.

**RNA-seq.** RNA-seq was performed by the Washington University Genome Technology Access Center (GTAC). Library preparation was performed with 10 μg of total RNA with a Bioanalyzer RIN score greater than 8.0. Ribosomal RNA was removed by poly-A selection using Oligo-dT beads (mRNA Direct kit, Life Technologies). mRNA was then fragmented in buffer containing 40 mM Tris acetate pH 8.2, 100 mM potassium acetate and 30 mM magnesium acetate and heating to 94 °C for 150 s. mRNA was reverse transcribed to yield cNDA using SuperScript III RT enzyme (Life Technologies, per manufacturer's instructions) and random hexamers. A second strand reaction was performed to yield ds-cDNA. cDNA was blunt ended, had an A base added to the 3′ ends, and then had Illumina sequencing adapters ligated to the ends. Ligated fragments were then amplified for 12 cycles using primers incorporating unique index tags. Fragments were sequenced on an Illumina HiSeq-3000 using single reads extending 50 bases.

RNA-seq reads were aligned to the Ensembl release 76 top-level assembly with STAR version 2.0.4b. Gene counts were derived from the number of uniquely aligned unambiguous reads by Subread:featureCount version 1.4.5. Transcript counts were produced by Sailfish version 0.6.3. Sequencing performance was assessed for total number of aligned reads, total number of uniquely aligned reads, genes and transcripts detected, ribosomal fraction known junction saturation and read distribution over known gene models with RSeQC version 2.3.

To enhance the biological interpretation of the large set of transcripts, grouping of genes/transcripts based on functional similarity was achieved using the R/Bioconductor packages GAGE and Pathview. GAGE and Pathview were also used to generate pathway maps on known signaling and metabolism pathways curated by KEGG.

**Targeted metabolomics.** We performed targeted metabolomics as reported with minor modifications [57]. Briefly, the liver samples were homogenized in water (4 mL/g liver). The amino acids in 20 μL of mouse serum or liver homogenate were extracted with protein precipitation in the presence of internal standards (13C6,15N-Ile, d3-Leu, d8-Lys, d8-Phe, d8-Trp, d4-Tyr, d8-Val, d7-Pro, 13C4-Thr, d3-Met, d2-Gly, d3,15N2-Asn, d4-Cit, d3-Asp, 13C5-Gln, 13C6-His, d3-Glu, d4-Ala, d3-Ser, 13C5-Orn, and 13C6-Arg). Quality control (QC) samples for livers and sera were prepared from pooled partial study samples and injected every five study samples to monitor intra-batch precision. Only the lipid species with CV% < 15% for QC injections are reported. The Ile, Leu, Lys, Phe, Trp, Tyr, Val, Pro, Thr, Met, Gly, Asn, Cit, Asp, Gln, His, Glu, Ala, Ser, Orn, and Arg were analyzed on 4000 QTRAP mass spectrometer coupled with a Prominence LC-20AD HPLC system. Data processing was conducted with Analyst 1.5.1 (Applied Biosystems).

**Statistical analyses.** Data were analyzed using GraphPad Prism version 6.0 (RRID: SCR_015807). $p < 0.05$ was defined as statistically significant. Data shown are as mean ± SEM. In box-whisker plots: middle bar represents the dataset median; boxes represent 25 and 75 percentile lines; whiskers represent maximum and minimum values in the dataset. In dot plots: the line represents the data mean. Unpaired two-tailed homoscedastic T-tests with Bonferroni post hoc correction for multiple comparisons were used for all analyses unless otherwise noted in the figure legends. Two-way ANOVA was also used for analyses with two independent variables.

**Reporting summary.** Further information on experimental design is available in the Nature Research Reporting Summary linked to this article.

## Data availability

There are no restrictions on data or material availability. The data that support the findings of this study are available from the authors on reasonable request. Raw RNA sequencing data files have been deposited into the Gene Expression Omnibus database (www.ncbi.nlm.nih.gov/geo/) with accession number GSE126134.

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

## Acknowledgements

We thank Mark Graham (IONIS Pharmaceuticals, Carlsbad, CA) for providing Arg2-specific antisense oligonucleotide. This work was supported by the Office of the Assistant Secretary of Defense for Health Affairs, through the Peer Reviewed Medical Research Program under Award No. W81XWH-17-1-0133. This work was also supported by grants from the NIH/National Center for Advancing Translational Sciences (NCATS) grant #UL1TR002345, the NIH R56 (DK115764), the Children's Discovery Institute (MI-FR-2014-426 and MI-II2017-593), AGA-Gilead Sciences Research Scholar Award in Liver Disease, the Washington University Digestive Disease Research Core Center (BJD) (P30DK52574), the Washington University Diabetes Research Center (P30DK020579), Nutrition & Obesity Research Center (P30DK056341), and the Robert Wood Johnson Foundation. B.M.S. was supported by National Institutes of Health (R15 AI117670). A.L.M. was supported by the Washington University Spencer T. Olin Fellowship, the Washington University NIGMS Institutional Training Grant in Cell and Molecular Biosciences (T32GM007067), and National Science Foundation Graduate Student Fellowship (DGE-1143954).

## Author contributions

B.J.D. conceived and coordinated the study and wrote the paper. Y.Z., C.B.H., H.F., and B.J.D. designed, performed and analyzed the experiments. A.L.M. analyzed the experiments. B.M.S. and A.I.S. designed and analyzed the lactotrehalose experiments and

designed and optimized lactotrehalose synthesis. All authors reviewed the results and approved the final version of the manuscript.

## Additional information

**Competing interests:** A provisional patent application, "AAV-mediated hepatocyte Arginase II overexpression as a treatment against metabolic disease" is pending, on which B.J.D. is the lead inventor. The data in Figs. 1–5 and 9–10 and Supplementary Figs. 1, 2, and 5 are pertinent to the filed application. The authors declare no competing interests.

