## [Peer Review File · Nature Communications]

Reviewers' comments:

Reviewer #1 (Remarks to the Author):

In the study by Zhang et al, has been reported for the first time the evidence of arginase 2 (Arg2) as a fasting-induced hepatocyte factor that protect against diet or genetically induced non-alcoholic fatty liver disease (NAFLD) traits (hepatic fat accumulation and inflammation), insulin- and glucose intolerance. The authors also demonstrated that the mechanism by which Arg2 regulate hepato-metabolism is dependent on G-protein signaling. All of these data paper represent an advance in understanding new molecular mechanisms that lead to NAFLD, defining a novel therapeutic effector pathway against hepato-metabolic effects associated to this obesity-driven disease.

The study is relevant for the field and the experiments are well conducted and I have found only few critical points that should be improved:

Results:

Arg2 is induced during..... This paragraph report some data that confirmed previous published data so are experimentally but not conceptually new. Thus I suggest to put some figures in supplementary (i.e. 1A, B and C). Moreover, it should be mentioned the concentration used for in vitro experiments and the motif of these concentrations that are not reported in results and legend but only on the histograms. The figure1J should reported also HFHS mice data as stated in the text.

Hepatic Arg2 attenuates.... I suggest to replace figure 1D in supplementary and made this panel more readable.... It report interesting results but marginal for the core of this paragraph. Panel 1F lacks of data about control-NCD and Arg2-NCD.

Hepatic Arg 2 enhances glucose.... and Hepatic Arg2 mitigates.... I suggest to incorporate these paragraphs in a unique. Figure 4F check for legends descriptors ITT....or GTT. Figure H, should report also Arg2 expression.

Figure 8F.... is not explicative is a simple summary that not had clear information to the manuscript that is more complex. I suggest to remove this scheme.

Minor points:

- Provide quantitative data of WB when possible and reported for at least 4 experiments.....it is crucial for ensuring repeatability.
- In general, check all figure by using the same characters and points....moreover made clear on the histograms when they reported mRNA or protein by using a clear name axis that should be the same for all histograms.
- Check for minor typos

Reviewer #2 (Remarks to the Author):

The authors seek to test the hypothesis that hepatic arginase 2 is a central factor in the adaptive

changes in hepatic and systemic metabolism by which calorie reduction and intermittent fasting protect against obesity-induced cardiovascular and metabolic disease. Further, they state that RGS reduction is a key event in this protective adaptation.

The manuscript is of interest but there are problems and concerns with the methods, data and the authors' interpretations. Use of more straight-forward words and phrases would also increase clarity.

Specific points:

1. Use of AAV serotype 8 does preferentially target the liver but it has affinity for a number of other tissues. Why was skeletal muscles and brown fat used to assess selectivity? What about delivery to visceral adipose tissue, in which Arg2 is elevated in obesity, or possibly kidney? Are Kupffer cells and hepatic stellate cells affected, as well as hepatocyte?
2. Does ARG2 over expression in liver affect enterohepatic portal circulation or intestinal food absorption?
3. Does overexpression ARG2 in liver affect ARG1 expression in liver? – there is evidence that overexpression or lack of one of the isoforms decreases or elevates the other.
4. This work was based on a finding that arginase 2 was upregulated when hepatic glucose uptake is inhibited (with trehalose). However, in type 2 diabetic patients, hepatic glucose uptake is reported to be impaired (Iozzo, P. et al, J Clin Endoc & Metab, 2003). Comments.
5. Western blots for ARG2 throughout the figures are not very clear and do not substantiate the stated alterations in ARG2 expression. See Figs 1 and 7. In Fig 7 blots, why is the background for Arg2 so much darker than other proteins? Weight markers are need for all WB images.
6. In the materials and methods sections the authors described that they isolated primary hepatocytes from WT mice, while they used AML12 cells for in vitro insulin signaling presented in Fig 1I, while the legend states that they used primary hepatocytes. Also in this Fig. the increase in ARG2 expression for the AAV8-arg2 group compared to control is not convincing
7. For using indirect calorimetry and food intake measurement, acclimation usually requires over 24 hrs before recording. A period of 1-3 hrs is not enough.
8. In Fig 2 D for respiratory exchange ratio [RER], AAV8 -Arg2 NCD exhibited significantly lower value than AAV8 control which indicates metabolic inflexibility and insulin resistance as discussed in <https://www.ncbi.nlm.nih.gov/pmc/articles/PMC1159159/>. However, insulin resistance as measured via HOMA2 is lower in AAV8 -Arg2 NCD group
9. Results section page 4: “hepatic Arg2 overexpression increased serum free fatty acid (FFA) in AAV8-Arg2 HFrD mice with no significant changes in serum triglyceride (TG), total cholesterol, and LDL-C (fig. S1D)” Increased free fatty acid levels have been reported to induce insulin resistance, hyperlipidemia, and diabetes.
10. Authors state that the effects of enhanced Arg2 levels are not due to increased enzyme activity. Their evidence is not convincing. Treatment with an arginase inhibitor would reveal whether enzymatic activity is involved, even though Arg1 activity would also be inhibited.
11. Supplementary Fig 1I shows that ARG2 HFrD significantly increased lipid synthesis gene expression which contradicts the lower hepatic triglycerides.
12. Figure characters are very difficult to read. Letters within figure panels are too small. In Fig 1J, which gene is involved?
13. Why did authors use two-tailed T-test rather than ANOVA?

Reviewer #3 (Remarks to the Author):

In this manuscript Zhang et al. describe a mechanism whereby Arg2 regulates Rgs16 signaling to modulate the hepatic glucose fasting response, and hepatic and extra-hepatic energy homeostasis. The authors demonstrate that Arg2 is induced after fasting, and negatively correlates with Rgs16 levels. Additionally, in diabetic mouse models, Arg2 levels are suppressed. The authors further show that hepatic Arg2 expression, in both high fructose diet-induced and db/db diabetic mouse models, enhances basal thermogenesis, and protects mice from weight gain, insulin resistance, glucose intolerance, hepatic steatosis and hepatic inflammatory cytokine/chemokine expression. The authors additionally claim that Arg2-regulated Rgs16 signaling contributes to the observed effects. While the concepts are interesting, the novel findings reported and the model hypothesized are not evaluated in sufficient depth mechanistically to add significantly to the understanding of the importance of Arg2 and Rgs16 in hepatic metabolism. The main issue is the rationale or why in the fasting response Arg2 controls Rgs16? Is this part of a glucagon response? But why this enzyme controls this signaling, and how is this integrated in the fasting response. Additionally, other concerns are mentioned below.

Major concerns:

1. Are the doses of trehalose used in Fig 1 comparable to the K_i (if known) for the GLUT8 receptor? Or would the doses used cause off-target effects?
2. The authors mention Sirt1 in Fig 1G, but not much is explored after this in terms of mechanistic importance of Sirt1, and hence the data appears superficial. If not explored further, this data should not be included.
3. In a number of graphs, while the authors indicate the changes are significant, they appear minimal, and hence it is hard to fully convince the reader that the changes have a significant downstream impact. For example the changes in RER (Dark) in Fig 2D, RER (Light) in Fig 4D and RER (Light) in Fig 7D, while significant, are minimal – and hence the question arises whether these changes are consequential for overall metabolism or not.
4. Most, if not all, of the in vivo experiments involve an overexpression of Arg2. The data would be strengthened if experiments were conducted with a liver-specific deletion of Arg2. Additionally, experiments involving siRNA-mediated knockdown of Arg2 in isolated hepatocytes could complement the data.
5. Quantification of some of the western blots would be useful, such as Fig 1I and Fig 4H.
6. For comparison reasons, it would be nice if the authors included data/blots for Arg2 in Fig 6F and 6G, and Rgs16 in Fig 7G.
7. Are there known downstream direct or indirect targets/roles of Arg2 and Rgs16? Are these regulated too upon fasting and in *ob/ob* mice?
8. The authors have tested the effects in hepatic tissue and hepatocytes – for in vivo experiments, it would be nice to see the effects of hepatic Arg2 overexpression on extra-hepatic tissues like the fat and muscle. For example is insulin sensitivity altered?
9. The authors claim a role for Arg2 in insulin resistance/sensitivity. This would be highly strengthened by testing the effects of Arg2 using a hyperinsulinemic-euglycemic clamp in vivo.

NCOMMS-18-15282A
RESPONSES TO REVIEWER COMMENTS

We are grateful to our Reviewers for their insightful comments. In response to these helpful suggestions for additional lines of experimentation, additional controls, and augmented data interpretation, we strengthened the manuscript's impact and conclusions.

To summarize major enhancements to the manuscript:

- We generated a new, liver-selective Arg2 deficiency model in collaboration with IONIS Pharmaceuticals using antisense oligonucleotide (ASO) technology (New Figs: 6, 7, S3 and S4).
- We interrogated the hepatic Arg2-deficient mouse model to evaluate the effects of hepatic Arg2 deficiency on thermic, lipid, and glucose-insulin homeostasis. We report the first demonstration of increased circulating insulin and HOMA2 IR indices for insulin resistance in the setting of hepatic Arg2 deficiency (Fig. 6, S3).
- We evaluated the effects of liver-specific Arg2 deficiency on hepatic inflammation, lipid accumulation, and gene expression profiling of *de novo* lipogenesis, gluconeogenesis, lipid import/efflux, β -oxidation, and insulin sensitivity. We demonstrate increased hepatic steatosis and inflammatory markers in context of hepatic Arg2 deficiency (Fig. 7; S3, S4).
- We confirmed hepatic Arg2 expression by demonstrating Arg2 enzymatic activity upon viral Arg2 overexpression. This complemented the Arg2 protein and mRNA-based demonstrations of Arg2 upregulation in the original manuscript (Letter Fig. 7B, Manuscript Fig. S1A, S2A, S2B).
- We performed additional controls for Arg2 specificity of overexpression in kidney and white adipose tissue, in addition to the initial brown adipose and skeletal muscle controls. We further demonstrated that the Arg1 isoform is not downregulated as a compensatory genetic change in any of our multiple models (Fig. S1A, S2A, S3A, S4).
- We amended and expanded the Discussion and refined figures and labeling for clarity in response to Reviewer suggestions.

Reviewer #1 (Remarks to the Author):

In the study by Zhang et al, has been reported for the first time the evidence of arginase 2 (Arg2) as a fasting-induced hepatocyte factor that protect against diet or genetically induced non-alcoholic fatty liver disease (NAFLD) traits (hepatic fat accumulation and inflammation), insulin- and glucose intolerance. The authors also demonstrated that the mechanism by which Arg2 regulate hepato-metabolism is dependent on G-protein signaling. All of these data paper represent an advance in understanding new molecular mechanisms that lead to NAFLD, defining a novel therapeutic effector pathway against hepato-metabolic effects associated to this obesity-driven disease. The study is relevant for the field and the experiments are well conducted and I have found only few critical points that should be improved.

Results:

Arg2 is induced during... This paragraph report some data that confirmed previous published data so are experimentally but not conceptually new. Thus I suggest to put some figures in supplementary (i.e. 1A, B and C). Moreover, it should be mentioned the concentration used for in vitro experiments and the motif of these concentrations that are not reported in results and legend but only on the histograms. The figure 1J should reported also HFHS mice data as stated in the text.

Because SIRT1 indeed is a well-characterized, canonical fasting response regulator, and because it largely recapitulated the advances demonstrated with trehalose and lactotrehalose, we agree with the First Reviewer and removed those data from the original manuscript Fig. 1G. Reviewer 3 concurred with the First Reviewer in removing these data.

To improve communication of the drug doses used in each experiment, we added drug dosages to the revised Results and Figure Legends, primarily to Fig. 1.

We would like to highlight the conceptual advances and clinical application of some of the experiments in Fig. 1. Fig. 1A. shows for the first time a pharmacological means by which to induce Arg2, using the hepatic glucose transporter inhibitor, trehalose. Fig. 1B demonstrates the trehalose effect on Arg2 expression depends upon the energetic deficit introduced by trehalose, because feeding pyruvate distal to the GLUT blocks trehalose induction. Fig. 1C demonstrates that the resistant trehalose analogue, lactoTre, activates Arg2 mRNA accumulation even more potently than trehalose. The data are particularly important, in light of recent data by Collins and Britton, et al (1,2), which suggest that trehalose is susceptible to degradation by gut and microbial trehalases. Because lactoTre resists trehalase-mediated cleavage, it introduces a means to circumvent the theoretical limitations of trehalose therapy. In light of these clinically relevant advances, we move to feature these data in Fig. 1.

“Hepatic Arg2 attenuates...” I suggest to replace figure 1D in supplementary and made this panel more readable.... It report interesting results but marginal for the core of this paragraph. Panel 2F lacks of data about control-NCD and Arg2-NCD.

We replaced Supp. Fig 1D and enlarged the panel to make it more readable.

Fig. 2F Control-NCD and Arg2-NCD data are now shown in fig. S1E. The data show no basal differences in glucose or insulin tolerance when comparing AAV8-Control vs. AAV8-Arg2 mice on NCD. We recapitulated Fig. S1E in Letter Fig. 1.

Letter Fig. 1. Hepatic Arg2 does not alter basal glucose or insulin homeostasis. Shown are glucose and insulin tolerance test data from mice overexpressing control vector or hepatocyte-specific Arg2. Shown at right are the areas under the tolerance test curve (AUC). n.s., not significantly different vs. control group.

Hepatic Arg 2 enhances glucose.... and Hepatic Arg2 mitigates.... I suggest to incorporate these paragraphs in a unique. Figure 4F check for legends descriptors ITT....or GTT. Figure H, should report also Arg2 expression.

Each paragraph referenced by the Reviewer occupies its own unique Figure:

-Hepatic Arg2 enhances glucose homeostasis and thermogenesis in db/db diabetic mice is referenced in Fig. 4.

-Hepatic Arg2 mitigates hepatic steatosis in db/db mice is referenced in Fig. 5.

The Fig. 4F legend was verified to identify GTT and ITT data appropriately.

Letter Fig. 2 Arg2 overexpression in db/db mouse liver. Shown are Arg2 immunoblot data and densitometric quantification of those data in liver from db/db mice injected with either control vector, or Arg2 overexpression vector.

We reported Arg2 expression in Fig. 4H by mRNA and protein for the mice analyzed now as fig S2B. We recapitulated fig. S2B here as Letter Fig. 2. It shows that AAV8-Arg2 treatment increases hepatic Arg2 protein.

Figure 8F.... is not explicative is a simple summary that not had clear information to the manuscript that is more complex. I suggest to remove this scheme.

We removed this working model of Arg2 action. We removed description of this model in the revised Discussion.

Minor points:

Provide quantitative data of WB when possible and reported for at least 4 experiments.....it is crucial for ensuring repeatability.

We quantified immunoblot data in the revised Manuscript Fig. 1H, 4H, 8F, 8G, 9G, S1A, and S2B.

In general, check all figure by using the same characters and points....moreover made clear on the histograms when they reported mRNA or protein by using a clear name axis that should be the same for all histograms.

We made uniform (and enlarged) all fonts, and clearly labeled our axes.

Check for minor typos

Thank you – we carefully reviewed the revised manuscript for typographical errors.

Reviewer #2 (Remarks to the Author):

The authors seek to test the hypothesis that hepatic arginase 2 is a central factor in the adaptive changes in hepatic and systemic metabolism by which calorie reduction and intermittent fasting protect against obesity-induced cardiovascular and metabolic disease. Further, they state that RGS reduction is a key event in this protective adaptation.

The manuscript is of interest but there are problems and concerns with the methods, data and the authors' interpretations. Use of more straight-forward words and phrases would also increase clarity.

Specific points:

1. Use of AAV serotype 8 does preferentially target the liver but it has affinity for a number of other tissues. Why was skeletal muscles and brown fat used to assess selectivity? What about delivery to visceral adipose tissue, in which Arg2 is elevated in obesity, or possibly kidney? Are Kupffer cells and hepatic stellate cells affected, as well as hepatocyte?

We achieved hepatic transgene specificity in two ways. First, our AAV8 vector delivered Arg2 driven by a liver-specific thyroxine binding globulin (TBG) promoter (3,4). This promoter does not significantly express in non-parenchymal cell types in the liver or in extrahepatic organs, as shown in Letter Fig. 3 (derived from Yan, et al *Gene* 2012 (3,4)). Second, to gain further specificity of tissue expression for our Arg2 transgene, we used an AAV8 vector, which is tropic to the liver parenchyma.

We analyzed Arg2 expression in several extrahepatic tissues, and showed similar specificity for Arg2 overexpression with our vector (Letter Fig. 4). Although we did not analyze kidney in our initial overexpression cohorts, we obtained and analyzed renal tissue in our new liver Arg2-deficient mice. We generated this new model in response to Reviewer suggestions for this manuscript revision. It is important to note, the liver Arg2-deficient model exhibited a phenotype that is congruent with that already demonstrated in our several overexpression models. Liver Arg2-deficient mice had exacerbated NAFLD and hyperinsulinemia, when compared with control *db/db* mice. again suggesting that hepatic Arg2 is protective against NAFLD. Yet, Arg2 was knocked down selectively in liver, not in kidney (Letter Fig. 4, grey and green bars). Kidney Arg2 expression data, as well as are added in the revised manuscript Results section (page 5, last paragraph).

Letter Fig. 3. Data from Yan, et al *Gene* 2012. Liver-specific, thyroxine binding globulin (TBG) promoter expression in a lentivirus vector, previously published. In the current manuscript, to gain further tissue- and cell-type specificity we used an AAV8 vector. **, $P < 0.01$ vs. all other tissues examined.

Letter Fig. 4. Arg2 overexpression interscapular brown adipose tissue (iBAT), skeletal muscle (SKM) and visceral white adipose tissue (WAT). Kidney Arg2 expression is shown after control or Arg2 ASO treatment. Basal liver Arg2 expression is set as the reference value. *, ****, $P < 0.05$, < 0.0001 vs. control. n.s., no significant difference.

We assayed skeletal muscle and brown adipose to assess selectivity of overexpression because ectopic Arg2 overexpression in these metabolically active tissues might have confounded our conclusion, that *hepatic* Arg2 is sufficient to convey the metabolic effects of fasting. We added (white) visceral adipose tissue Arg2

analyses in our overexpression models in the revised manuscript fig. S2A (also here in Letter Fig. 4). This revealed no Arg2 overexpression in any extrahepatic tissues, including visceral adipose tissue (Letter Fig. 4).

2. Does ARG2 over expression in liver affect enterohepatic portal circulation or intestinal food absorption?

Although we did not directly measure portal blood flow, Arg2OE mice did not exhibit the signs or symptoms of altered portal circulation. Spleen size and abdominal wall vasculature were grossly normal in Arg2OE mice. No abnormalities in bleeding time or platelet counts were observed to suggest hyper- or hyposplenism or thrombocytopenia.

We do not observe evidence of malabsorption. The decreased weight gain that we observed in Arg2-overexpressing (Arg2OE) mice could be secondary to increased caloric expenditure, caloric food intake, or malabsorption. Caloric intake and it is not decreased in Arg2OE mice, whereas caloric expenditure is increased in Arg2OE mice (Fig. 2D, 4D). Finally, we did not observe clinical signs and symptoms of malabsorption: diarrhea, increased stool output and/or steatorrhea. In our metabolic cage experiments, stool volume, character and consistency did not change between groups. On the basis of these data, we postulate that increased caloric expenditure could be sufficient to account for the decreased weight gain in Arg2-overexpressing mice, although our current data would not completely rule out a malabsorptive phenotype.

3. Does overexpression ARG2 in liver affect ARG1 expression in liver? – there is evidence that overexpression or lack of one of the isoforms decreases or elevates the other.

Letter Fig. 5. No change in hepatic Arg1 expression upon Arg2 overexpression. Shown is Arg1 quantification by in livers from WT mice (left) and db/db mice (right) on chow or HFrD. **, ****, $P < 0.01$, $P < 0.0001$ vs. control. n.s., not significantly different vs. control.

Letter Fig. 6. No change in hepatic Arg1 expression upon hepatic Arg2 knockdown. Shown is Arg1 (left) and Arg2 (right) mRNA quantification in livers from db/db mice treated with Control ASO or Arg2 ASO. ****, $P < 0.0001$ vs. control. n.s., not significantly different vs. control.

Hepatic Arg2 overexpression did not suppress hepatic Arg1 expression, in WT mice fed chow or high-fructose diet, or in db/db mice (Letter Fig. 5). We also measured Arg1 in liver tissue from mice treated with Arg2 antisense oligonucleotide. Arg2 knockdown did not compensatorially upregulated hepatic Arg1 mRNA expression (Letter Fig. 6). Interestingly, we observed a 10% increase in hepatic Arg1 mRNA expression uniquely in db/db AAV8-Arg2 mice, versus a 20-fold increase in hepatic Arg2 in AAV8-Arg2 mice (Letter Fig. 5). Even though hepatic Arg1 did not change in any of our other Arg2 overexpression or knockdown models, we observed congruent phenotypes. That is, Arg2 acted as a protective factor in every model examined, irrespective of hepatic Arg1 expression. Arg1 expression data are now presented as fig. S1A, S2A and S3A.

4. This work was based on a finding that arginase 2 was upregulated when hepatic glucose uptake is inhibited (with trehalose). However, in type 2 diabetic patients, hepatic glucose uptake is reported to be impaired (Iozzo, P. et al, J Clin Endoc & Metab, 2003). Comments.

This is a fascinating observation. The apparent paradox is resolved by highlighting the differences between the diabetic and the fasted state. Ultimately these differences underlie part of the impact of this paper. At least two key observations resolve the paradox. First, insulin stimulates hepatic glucose uptake by 2-fold over the fasting state. In diabetic patients, even though insulin-stimulated glucose uptake is impaired (by 17.4% in the Iozzo study), this blunted fed-state response would not be expected to mimic a true hepatocyte fasting response, because hepatocyte glucose influx is still a net increase in the diabetic vs. fasting state.

Second, apart from insulin-stimulated glucose uptake, basal hepatic glucose uptake should remain elevated through facilitative transporters GLUT1, GLUT2 and GLUT8. This is because fasting and fed hyperglycemia in the diabetic state provides a concentration gradient that drives these facilitative transporters even in the absence of insulin stimulated glucose transport. This rise in basal hepatic glucose uptake should further impair the hepatocyte's ability to achieve a true fasting response.

We added this discussion to the revised manuscript Discussion session (page 7, last paragraph)

“The observation that hepatic Arg2 activation is therapeutic originated from prior observations that hepatic fasting signals (e.g. TFEβ, PPARα) mediated the therapeutic effects of genetic or pharmacological glucose transport blockade (5,6). However, the observation that hepatic glucose uptake is impaired in diabetic patients (7) may raise an apparent paradox, in light of the in-

trinsic therapeutic effect of hepatic glucose transport blockade. At least two key differences between the fasted liver and the diabetic liver resolve this apparent paradox. First, insulin stimulates hepatic glucose uptake by 2-fold over the fasting state (8). In diabetic patients, insulin-stimulated glucose uptake is impaired (by 17.4% in the Iozzo study (7)), but this degree of blunted fed-state glucose uptake would not be expected to mimic a true hepatocyte fasting response. Second, *basal* hepatic glucose uptake should remain elevated through hepatic facilitative transporters GLUT1, GLUT2 and GLUT8. This is because basal hyperglycemia in the diabetic patient provides a glucose concentration gradient that drives these facilitative transporters even in the absence of insulin-stimulated glucose transport. This rise in basal hepatic glucose uptake would be expected to further impede the hepatocyte fasting response. Thus, although insulin-stimulated glucose uptake is relatively impaired in the insulin-resistant state, our data suggest that pharmacological hepatic GLUT blockade or targeted hepatic Arg2 activation can drive the therapeutic hepatic glucose response in diet-induced and genetic models of obesity and insulin resistance.”

5. Western blots for ARG2 throughout the figures are not very clear and do not substantiate the stated alterations in ARG2 expression. See Figs 1 and 7. In Fig 7 blots, why is the background for Arg2 so much darker than other proteins? Weight markers are need for all WB images.

Thank you, we added weight markers to all immunoblot images in the manuscript.

We substantiated Arg2 overexpression first immunoblot, and also using two additional methods: qPCR and Arg2 enzymatic activity (Letter Fig 7). Depending on tissue type, protein concentration in the lysate, etc, we acknowledge that the antibody (SCBT #AB154422) can give suboptimal results. Importantly, we analyzed five different antibody sources to find the current antibody (2 Abcam antibodies, 2 SCBT, and 1 Cell Signaling Technologies antibody), which we then optimized extensively. Nevertheless, we acknowledge the antibody’s limitations. The background (e.g. multiple bands observed by immunoblot) observed is likely to be at least in part due to Arg2 O-glcNAcylation, which was recently demonstrated (9).

To further interrogate the degree of Arg2 expression in our models, we quantified Arg2 expression by qPCR analysis using our vectors in hepatocyte cultures and *in vivo*. We demonstrated 10-30-fold increases in Arg2 mRNA in Arg2-transfected liver and hepatocyte cultures (Letter Fig. 7A). To demonstrate that our construct expressed enzymatically functional protein, we overexpressed Arg2 *in vitro*, and measured Arg2 urea production (Letter Fig. 7B and revised manuscript Results section, pg. 3, second paragraph). Arg2OE hepatocytes had 2.5-fold greater urea production. Our immunoblot data and quantification of these data are recapitulated in Letter Fig. C for reference. Together, we interpret gene expression, immunoblot data and enzymatic activity to suggest that our overexpression vectors achieve Arg2 overexpression of mRNA and enzymatically functional protein.

Letter Fig. 7. Verification of Arg2 overexpression by (A) Liver qPCR, (B) Arginase activity in AML12 hepatocytes, and (C) immunoblot analysis in livers overexpressing control or Arg2 overexpression vector. *, **, **** $p < 0.05$, < 0.01 , < 0.0001 vs. control. n.s., not significantly different vs. control.

6. In the materials and methods sections the authors described that they isolated primary hepatocytes from WT mice, while they used AML12 cells for in vitro insulin signaling presented in Fig 1I, while the legend states that they used primary hepatocytes. Also in this Fig. the increase in ARG2 expression for the AAV8-arg2 group compared to control is not convincing.

Both AML12 and primary hepatocytes were used for these experiments, as we have done previously (5,10). We clarified cell types used in each of our cell culture experiments in the revised manuscript Figure Legends. Specifically, Figure 1L used AML12 cells. We acknowledge the limitations of the antibody, and so in light of the difficulty with immunoreactive band imaging, we measured ureagenesis activity in AML12 cells overexpressing Arg2 in parallel. We showed that Arg2 overexpression increased ureagenesis by ~2.5-fold (Letter Fig. 7B). Thus, mRNA, protein and enzymatic activity support the premise that Arg2 is overexpressed in the models presented.

7. For using indirect calorimetry and food intake measurement, acclimation usually requires over 24 hrs before recording. A period of 1-3 hrs is not enough.

Acclimation periods in the published literature range from 2hrs (11) to 4 hours (6,10) to 72hrs. To our surprise, we found that a number of publications do not report acclimation time at all. We optimized our own indirect calorimetry assays in order to define the earliest time points at which caloric expenditure stabilizes. efficiency step allowed us to collect the largest amount of calorimetric data from each animal while also allowing us to increase the number of different animals analyzed per experiment. This was necessary, in view of quite limited calorimetry availability at our center.

Pilot and optimization data are shown in Letter Fig. 8. We quantified mean heat generation for during hour #4 and during hour #28 (e.g. heat generation after 3 or after 27 hr acclimatization). This full 24hr shift controlled for the absolute time of day at which the calorimetric recordings were made. As we report throughout the manuscript, Arg2-overexpressing mice exhibited increased caloric consumption, and oxygen - carbon dioxide exchange both after 3hr acclimation and after 24 + 3 hrs acclimation (Letter Fig. 8). In addition, when comparing heat generation within each group of mice after early vs. late acclimation, there was no difference in heat generation ($p = 0.11$ for AAV8 Controls and $p = 0.359$ for AAV8 Arg2OE mice). We verified identical results across all our calorimetry data. In order to standardize all data to identical acclimation times, we analyzed all data as strictly after 4hr acclimation, and noted this in the revised manuscript Methods section (pg. 11, first paragraph). This requantification did not change any of the statistical results or conclusions stated in the manuscript.

8. In Fig 2 D for respiratory exchange ratio [RER], AAV8 -Arg2 NCD exhibited significantly lower value than AAV8 control which indicates metabolic inflexibility and insulin resistance as discussed in <https://www.ncbi.nlm.nih.gov/pmc/articles/PMC1159159/>. However, insulin resistance as measured via HOMA2 is lower in AAV8 -Arg2 NCD group

As a net average over multiple tissue types, a lower RER indicates an organism-wide predilection for fat oxidation. This is an expected RER response in an organism that is fasting (12), as glucose metabolism gives way to fat oxidation. Lower RER in Arg2OE mice supports the hypothesis that Arg2OE, in some parameters, mimics a fasting-like state.

Letter Fig. 8. Indirect calorimetry after 3 or 24hr + 3hr acclimation in the metabolic cage, analyzed in mice treated with AAV8 control or AAV8 Arg2. *, $P < 0.05$, vs. control during either time period.

The metabolic flexibility discussed by Dr. Kelley, (*J Clin Invest* 2005; PMC 1159159) referred to the transition of from fat to glucose oxidation between fed and fasting states. Because we quantified the *average* light- and dark-cycle- RER, and we did not specifically sub-stratify fasting and post-prandial RER within cycles, we are more cautious about drawing conclusions specifically regarding metabolic flexibility *per se*. Assessing whether hepatic Arg2 influences metabolic flexibility would indeed be an important avenue for future investigation. Nevertheless, as the Reviewer notes, hepatic Arg2 in this model appears to enhance insulin sensitivity.

9. Results section page 4: “hepatic Arg2 overexpression increased serum free fatty acid (FFA) in AAV8-Arg2 HFrD mice with no significant changes in serum triglyceride (TG), total cholesterol, and LDL-C (fig. S1D)”..... Increased free fatty acid levels have been reported to induce insulin resistance, hyperlipidemia, and diabetes.

The canonical response to fasting is to mobilize FFA from peripheral adipose stores via triacylglycerol lipolysis (13,14). The primary purpose of this is to provide substrate for fatty acid oxidation and gluconeogenesis in the liver. In the obese diabetic state, the context surrounding increased FFA is much different than in the fasted state. Increased FFA is associated with increased dietary fat intake in context of impaired mitochondrial fat oxidation is suppressed in the liver (15,16). It thus appears that the etiology, context, and fate of the circulating FFAs dictate in large part the FFA downstream effects. Arg2OE mice have a fat oxidation predilection (e.g. lower RER), lower peripheral and hepatic fat mass, and enhanced insulin sensitivity. The data from the Arg2OE model are most consistent with a pseudo-fasting, lipolytic, fat oxidative state.

10. Authors state that the effects of enhanced Arg2 levels are not due to increased enzyme activity. Their evidence is not convincing. Treatment with an arginase inhibitor would reveal whether enzymatic activity is involved, even though Arg1 activity would also be inhibited.

We agree that there are not yet enough data to draw conclusions regarding the role of Arg2 enzymatic activity *per se* in mediating the therapeutic effects of Arg2 overexpression. Indeed, we showed that Arg2 increases hepatocyte ureagenesis (recapitulated in Letter Fig. 9). Therefore, we withdrew discussion on the role Arg2 enzymatic activity in our observations. Instead, we offered a conservative interpretation of the data until more data are available. We **omitted** the following paragraph regarding ureahydrolase activity:

“Regarding Arg2 enzymatic function, GC-MS characterization of hepatocyte lysates suggests that changes in the amino acid pool due to arginase 2 ureahydrolase activity are not likely to explain the observed metabolic enhancements in Arg2-overexpressing mice. Indeed, we observed no significant changes in upstream urea cycle substrates: citrulline and aspartate; or in direct arginase reactants: ornithine and arginine. Consistent with our findings, *in vivo* expression of ureahydrolase-deficient Arg2 did not alter the effect of Arg2 on mTOR and AMPK signaling in macrophages (17,18). Our data and others’ indicate that Arg2 alters hepatic inflammatory and insulin signaling as well as lipid metabolism independent of changes in urea cycle intermediates.”

11. Supplementary Fig 1l shows that ARG2 HFrD significantly increased lipid synthesis gene expression which contradicts the lower hepatic triglycerides.

We observed lower net hepatic TG in AAV8-Arg2 mice after HFrD feeding and in *db/db* AAV8-Arg2 mice. Lower TG accumulation can be due to the net sum of changes in any (or all) of the following four broad processes: lipid uptake into the hepatocyte, lipid synthesis, lipid export, and lipid consumption (e.g. oxidation, lipophagy, etc.). As the Reviewer notes, we observe hallmarks of increased lipid synthesis gene expression in Arg2OE

Letter Fig. 9. Arg2 overexpression increases hepatocyte ureagenesis. Hepatocyte cultures were subjected to forced expression of β -galactosidase or Arg2 overexpression. Urea production was quantified in the presence or absence of arginine substrate. **, ****, $P < 0.01$, $P < 0.0001$ vs. bracketed control group.

mice. Also, we observe enhanced expression of the lipid export gene MTTP, and decreased expression of the lipid uptake protein, CD36. Moreover, *db/db* mice distinctly downregulate lipid synthetic gene expression upon Arg2 overexpression. We interpret this to overall to mean that the lipid synthetic program is a relatively minor contributor to the overall lower TG accumulation phenotype observed in Arg2-overexpressing liver.

12. Figure characters are very difficult to read. Letters within figure panels are too small. In Fig 1J, which gene is involved?

Thank you, we enlarged the font in all figures. Figure 1J (Now Fig. 1I in the Revised Manuscript) shows decreased hepatic Arg2 expression in fructose-treated primary hepatocyte cultures, and in *db/db* mice. We recapitulated these data in Letter Fig. 10 for reference.

Letter Fig. 10. Arg2 is decreased in models of caloric excess. Shown is Arg2 expression in hepatocyte cultures fed regular media or 10mM fructose in regular media (left), or in *db/+* mice versus *db/db* obese, diabetic littermate mice. *, **, $P < 0.05$ and $P < 0.01$ vs. control

13. Why did authors use two-tailed T-test rather than ANOVA?

We chose two-tailed T-test because this statistical method is appropriate when directly comparing two groups. Examples of this in our study include comparing *db/db* Arg2 vs. *db/db* Control, or *db/db* Control ASO vs. *db/db* Arg2 ASO). Two-tailed T-test is also appropriate if multiple pair-wise comparisons are made with *post hoc* correction to account for the increased risk of falsely rejecting the null hypothesis when performing multiple comparisons. An example of this in our study includes *db/db* AAV8-Control vs. *db/db* AAV8-Arg2 vs. *db/db* AAV8-Arg2 Ad-RGS16. This statistical method has been reported previously (e.g. Gomez-Herreros et al *Nat Comm* 2017; Wang, Wander, Yuan et al *Nat Comm* 2017 (19,20)).

ANOVA tests the null hypothesis that any of three or more group means being compared are not different. If the null hypothesis is rejected, then essentially multiple pair-wise T-tests are performed to determine specifically which groups differ. Because the initial test on the dataset is first performed, ANOVA can in specific cases be less sensitive than directly performing T-tests with *post hoc* correction. We therefore elected to perform multiple T-tests and corrected by Bonferroni-Dunn *post hoc* correction.

Reviewer #3 (Remarks to the Author):

In this manuscript Zhang et al. describe a mechanism whereby Arg2 regulates Rgs16 signaling to modulate the hepatic glucose fasting response, and hepatic and extra-hepatic energy homeostasis. The authors demonstrate that Arg2 is induced after fasting, and negatively correlates with Rgs16 levels. Additionally, in diabetic mouse models, Arg2 levels are suppressed. The authors further show that hepatic Arg2 expression, in both high fructose diet-induced and *db/db* diabetic mouse models, enhances basal thermogenesis, and protects mice from weight gain, insulin resistance, glucose intolerance, hepatic steatosis and hepatic inflammatory cytokine/chemokine expression. The authors additionally claim that Arg2-regulated Rgs16 signaling contributes to the observed effects. While the concepts are interesting, the novel findings reported and the model hypothesized are not evaluated in sufficient depth mechanistically to add significantly to the understanding of the importance of Arg2 and Rgs16 in hepatic metabolism. The main issue is the rationale or why in the fasting response Arg2 controls Rgs16? Is this part of a glucagon response? But why this enzyme controls this signaling, and how is this integrated in the fasting response. Additionally, other concerns are mentioned below.

Major concerns:

1. Are the doses of trehalose used in Fig 1 comparable to the K_i (if known) for the GLUT8 receptor? Or would the doses used cause off-target effects?

Trehalose blocks total hepatocyte glucose transport by approximately 50-70% at 100mM in primary hepatocytes and HepG2 hepatocytes, respectively (21). We reported the IC_{50} for trehalose against the GLUTs to range from 17-126mM (21). At any given concentration, we expect that we are blocking multiple GLUTs. This is advantageous, since our primary “target” to induce Arg2 is glucose flux into the cell, regardless of the GLUT isoform mediating glucose entry. Nevertheless, trehalose was primarily used as a tool to interrogate effects of broad-spectrum GLUT inhibition in order to activate hepatocyte Arg2.

Letter Fig. 11. The glucose transport inhibitor trehalose induces hepatocyte Arg2. Shown is qPCR analysis of Arg2 in primary murine hepatocytes treated with 1, 10, or 100mM trehalose.

2. The authors mention Sirt1 in Fig 1G, but not much is explored after this in terms of mechanistic importance of Sirt1, and hence the data appears superficial. If not explored further, this data should not be included.

Thank you - we removed the former Fig. 1G. from the revised manuscript, as well as references to the data in the revised Abstract, Introduction, Results, Discussion, Figure Legends, and References.

3. In a number of graphs, while the authors indicate the changes are significant, they appear minimal, and hence it is hard to fully convince the reader that the changes have a significant downstream impact. For example the changes in RER (Dark) in Fig 2D, RER (Light) in Fig 4D and RER (Light) in Fig 7D, while significant, are minimal – and hence the question arises whether these changes are consequential for overall metabolism or not.

We agree with the Reviewer - the absolute magnitude of changes in a subset of our assays is modest. Overall, we did not evaluate the impact of any *single* metabolic change (for example, the specific, isolated impact of changes in RER) in Arg2OE mice. Taken together, however, we observed reduced hepatic steatosis, hepatic inflammation, body fat and body weight, as well as enhanced glucose tolerance and insulin sensitivity in several Arg2 models that are associated with relatively modest changes in RER, some gene expression, etc.

Depending on the measured parameter, modest changes can indeed have significant downstream impact in other contexts. For example, in humans, modest weight loss (between 5-10% initial weight) is sufficient to significantly reduce multiple cardiovascular risk factors in diabetic patients (22). This included improvements in HbA1c, blood pressure and lipid profiles. Together, the impact of our data speaks to the sufficiency of hepatic Arg2 to convey the broad therapeutic effects of generalized macronutrient fasting across multiple metabolic parameters in genetically diabetic and diet-induced obese models.

4. Most, if not all, of the in vivo experiments involve an overexpression of Arg2. The data would be strengthened if experiments were conducted with a liver-specific deletion of Arg2. Additionally, experiments involving siRNA-mediated knockdown of Arg2 in isolated hepatocytes could complement the data.

We concur that loss-of-function data will complement and further enhance the current data set. Therefore, we performed extensive metabolic characterization in an Arg2 loss-of-function model using a liver-selective antisense oligonucleotide (ASO, (5,6,10)) in collaboration with IONIS Pharmaceuticals (Carlsbad, CA).

Arg2 ASO knocked down hepatic Arg2 mRNA without altering Arg1 expression. (Letter Fig.12). Consistent with our in vivo Arg2 overexpression studies, hepatic Arg2 knockdown increased RGS16 mRNA in *db/db* Arg2 ASO mice. Also concordant with our overexpression model, hepatic Arg2 knockdown increased markers of hepatocellular damage, serum ALT and AST in *db/db* mice. Hepatic Arg2 knockdown also subtly increased total serum total cholesterol, whereas LDL-C increased by ~50% in *db/db* Arg2 ASO mice when compared with *db/db* mice treated with control ASO ($p < 0.0001$). Liver weight-to-body-weight ratio, hepatic TG, hepatic cholesterol and hepatic LDL-C were all significantly increased in *db/db* mice with hepatic Arg2 knockdown. In addition, hepatic Arg2 knockdown increased inflammatory markers IL-6, TNF α , CCL2 and CXCL9 between two- and ten-fold relative to *db/db* controls. Surprisingly, hepatic Arg2 knockdown did not exacerbate glucose or insulin intolerance in *db/db* mice (not shown). Taken together with the original manuscript's data, we conclude that hepatic Arg2 is both sufficient and necessary to protect against hepatic and circulating lipid dyslipidemia in diabetic mice.

Letter Fig. 12. Arg2 knockdown increases RGS16 without effects on Arg1. Shown are qPCR data measuring Arg1 (left), Arg2 (middle) and RGS16 (right) in livers from *db/db* mice treated with scrambled or Arg2-specific ASO. n.s, not significantly different.. *, ****, $P < 0.05$, < 0.0001 vs. control.

The clinical implications here are quite profound, especially in view of recent momentum toward inhibiting arginases in the setting of cardiovascular disease (23). Our data suggest that blocking Arg2 without sufficient specificity may produce negative metabolic consequences if hepatic Arg2 is inadvertently targeted. We added these data in the Revised Manuscript Figure 6, 7, S3 and S4, and added this discussion to the revised manuscript (Discussion, pg. 8, first paragraph).

Serum Enzyme and Lipids

Letter Fig. 13. Hepatic Arg2 knockdown increases serum transaminases and serum lipids. Shown is serum quantitation of ALT/AST activity and serum TG and LDL-C in *db/db* mice treated with scrambled or Arg2-specific ASO. n.s, not significantly different.. *, **, ****, $P < 0.05$, < 0.01 , < 0.0001 vs. control.

Hepatic Lipids

Letter Fig. 14. Hepatic Arg2 knockdown increases hepatic lipid accumulation. Shown is quantification of TG, cholesterol, LDL-C and FFA in hepatic lipid extract from *db/db* mice treated with scrambled or Arg2-specific ASO. n.s, not significantly different.. **, ****, $P < 0.01$, < 0.0001 vs. control.

Hepatic Inflammation

Letter Fig. 15 Hepatic Arg2 knockdown increases hepatic inflammatory marker gene expression. Shown are qPCR data measuring cytokine and chemokine mRNA in livers from *db/db* mice treated with scrambled or Arg2-specific ASO. *, ***, $P < 0.05$, < 0.001 vs. control.

5. Quantification of some of the western blots would be useful, such as Fig 1I and Fig 4H.

We quantified Fig. 1I and 4H from the original manuscript. In the revised manuscript, these are now Fig. 1H (shown here as Letter Fig. 17) and 4H (shown here in Letter Fig. 16). This revealed significantly increased AKT phosphorylation in *db/db* AAV8-Arg2 livers versus control livers, and in serum-starved hepatocytes overexpressing Arg2 after acute insulin treatment. Taken together, the data suggest that hepatocyte Arg2 enhances insulin signaling via the Akt pathway.

In addition to quantifying Fig. 1L and 4H, we quantified blots in the revised manuscript Fig. 8F, 8G, 9G, S1A, S2B. These showed: Significant suppression of RGS16 in Arg2-overexpressing liver from WT mice (Fig. 8F); and *db/db* mice (Fig. 8G); enhanced Akt phosphorylation in Arg2-overexpressing *db/db* livers, and reversal of this increased Akt phosphorylation in Arg2-overexpressing *db/db* livers with genetic RGS16 reconstitution (Fig. 9G); increased Arg2 protein in livers of WT mice (Fig. S1A) and *db/db* mice (Fig. S2B) treated with AAV8-Arg2 under control of the liver-specific thyroxine binding globulin (TBG) promoter.

6. For comparison reasons, it would be nice if the authors included data/blots for Arg2 in Fig 6F and 6G, and Rgs16 in Fig 7G.

We quantified Arg2 in the experiments from Figs. 6F, 6G (now corresponding to the revised manuscript Figs. 8F and 8G) in Figs. S1A and S2B (Letter Fig. 18). and RGS16 in Fig. 7G. This showed that WT and *db/db* Arg2 overexpressing livers have significantly lower RGS16 protein abundance.

Letter Fig. 16. Increased Akt phosphorylation in Arg2 overexpressing mouse liver. Shown is immunoblot quantification of phospho-AKT (S473) in livers from *db/db* mice treated with scrambled or Arg2-specific ASO. *, P < 0.05 vs. control.

Letter Fig. 17. Hepatic Arg2 knock-down exacerbates hepatic inflammation. Shown is qPCR quantification of inflammatory markers in livers from *db/db* mice treated with scrambled or Arg2-specific ASO. **, ****, P < 0.01, 0.0001 vs. control.

Letter Fig. 18. Hepatic AAV8-Arg2 increases Arg2 protein in WT and *db/db* mouse liver. Shown are immunoblot and densitometric quantification of Arg2 in livers from WT (left) and *db/db* mice (right) treated with Control or Arg2-encoding AAV8 under liver-specific TBG promoter control. *, P < 0.05 vs. control. These data are in the revised manuscript Fig. S1A and S2B.

We also showed data and blots for RGS16 from the original manuscript Fig. 7G (now revised manuscript Fig. 9G and Fig. S3B) are recapitulated here in Letter Fig. 19.

Letter Fig. 19. RGS16 adenovirus infection increases RGS16 protein and mRNA in db/db mouse liver. Shown are immunoblot and qPCR quantification of RGS16 in livers from WT (left) and db/db mice (right) treated with Control or RGS16-encoding adenovirus. *, P < 0.05 vs. bracketed comparison group. These data are in the revised manuscript Fig. 9G and ...

7. Are there known downstream direct or indirect targets/roles of Arg2 and Rgs16? Are these regulated too upon fasting and in ob/ob mice?

There are several parallel phenotypes between the fasted state and Arg2 expression. We postulate that the primary impact of the present study is the demonstration that Arg2 is sufficient to convey several of the therapeutic metabolic effects of fasting.

Pashkov et al (*J Biol Chem* 2011) showed that fasting suppresses RGS16, whereas RGS16 is induced in the fed state. Specifically, RGS16 is induced by *carbohydrate* feeding.

In the present study, fasting and hepatic GLUT blockade by trehalose treatment each induced Arg2 expression. Direct Arg2 overexpression suppressed RGS16, whereas, hepatic Arg2 knockdown significantly induced RGS16. Therefore, Arg2 is itself sufficient to regulate RGS16, independent of carbohydrate status.

In this manuscript, we also identified the downstream target pathways activated by Arg2 overexpression. We show that key pathways regulated by Arg2 are common to those regulated by fasting-regulated pathways (24,25). For example, fasting blocks inflammatory responses to obesigenic stimuli (24,26), and hepatic Arg2 overexpression recapitulated this inflammatory suppression (Fig. 1G, 3C, 5F, 8C). Fasting also suppresses ChREBP and glucokinase expression (27), and Arg2 overexpression recapitulated these findings in *db/db* mice (Fig. 5D, Fig. S5F, and Fig. 2G, 4G).

8. The authors have tested the effects in hepatic tissue and hepatocytes – for in vivo experiments, it would be nice to see the effects of hepatic Arg2 overexpression on extra-hepatic tissues like the fat and muscle. For example is insulin sensitivity altered?

We showed the effects of hepatic Arg2 overexpression on iBAT, SKM and WAT Akt phosphorylation as a surrogate for extra-hepatic insulin sensitivity. Quantifying these blots demonstrated trends toward increased Akt phosphoryla-

Letter Fig. 20. Enhanced AKT phosphorylation in extrahepatic tissue of db/db mice expressing hepatic Arg2. Shown are phospho-Akt (S473) immunoblots in extrahepatic tissues from mice expressing control vector or Arg2.

tion in WAT, SKM and iBAT.

9. The authors claim a role for Arg2 in insulin resistance/sensitivity. This would be highly strengthened by testing the effects of Arg2 using a hyperinsulinemic-euglycemic clamp in vivo.

Thank you - we agree to the value of this investigation as a future direction. In light of our HOMA2, in vivo insulin and glucose tolerance data, and hepatic in vitro and in vivo insulin signaling data, we rather focused our efforts on the hepatic Arg2 knockdown data to add the largest possible new dimension to the manuscript.

REFERENCES FOR REVIEWER RESPONSES

1. Collins J, Robinson C, Danhof H, Knetsch CW, van Leeuwen HC, Lawley TD, et al. Dietary trehalose enhances virulence of epidemic *Clostridium difficile*. *Nature* [Internet]. 2018; Available from: <http://www.nature.com/doi/10.1038/nature25178>
2. Collins J, Danhof H, Britton RA. The role of trehalose in the global spread of epidemic *Clostridium difficile*. *Gut Microbes* [Internet]. 2018;00(00):submitted. Available from: <https://doi.org/10.1080/19490976.2018.1491266>
3. Hayashi Y, Mori Y, Janssen OE, Sunthorntheeparakul T, Weiss RE, Takeda K, et al. Human Thyroxine-Binding Globulin Gene: Complete Sequence and Transcriptional Regulation. *Mol Endocrinol* [Internet]. 1993;7(8):1049–60. Available from: <http://dx.doi.org/10.1210/mend.7.8.8232304>
4. Yan Z, Yan H, Ou H. Human thyroxine binding globulin (TBG) promoter directs efficient and sustaining transgene expression in liver-specific pattern. *Gene* [Internet]. 2012;506(2):289–94. Available from: <http://dx.doi.org/10.1016/j.gene.2012.07.009>
5. Zhang Y, Higgins CB, Mayer AL, Mysorekar IU, Razani B, Graham MJ, et al. TFEB-dependent induction of thermogenesis by the hepatocyte SLC2A inhibitor trehalose. *Autophagy* [Internet]. 2018 Nov 2;14(11):1959–75. Available from: <https://doi.org/10.1080/15548627.2018.1493044>
6. Mayer AL, Zhang Y, Feng EH, Higgins CB, Adenekan O, Pietka TA, et al. Enhanced Hepatic PPAR α Activity Links GLUT8 Deficiency to Augmented Peripheral Fasting Responses in Male Mice. *Endocrinology* [Internet]. 2018;159(May):2110–26. Available from: https://watermark.silverchair.com/en.2017-03150.pdf?token=AQECAHi208BE49Ooan9kKhW_Ercy7Dm3ZL_9Cf3qfKAc485ysgAAAclwggG-BgkqhkiG9w0BBwagggGvMIIBqWIBADCCAaQGCSqGSIb3DQEHATAeBglghkgBZQMEAS4wEQQMqiPncCO-O1S5orLAgEQgIBdXix8E_mMHZxjdHhZcRa3VSwJj6VqSXSILZ7pRjmb
7. Iozzo P, Hallsten K, Oikonen V, Virtanen KA, Kemppainen J, Solin O, et al. Insulin-Mediated Hepatic Glucose Uptake Is Impaired in Type 2 Diabetes: Evidence for a Relationship with Glycemic Control. *J Clin Endocrinol Metab* [Internet]. 2003;88(5):2055–60. Available from: <https://academic.oup.com/jcem/article-lookup/doi/10.1210/jc.2002-021446>
8. Honka M-J, Latva-Rasku A, Bucci M, Virtanen KA, Hannukainen JC, Kalliokoski KK, et al. Insulin-stimulated glucose uptake in skeletal muscle, adipose tissue and liver: a positron emission tomography study. *Eur J Endocrinol* [Internet]. 2018 May 7;178(5):523–31. Available from: <http://www.ncbi.nlm.nih.gov/pmc/articles/PMC5920018/>
9. Aguilar H, Fricovsky E, Ihm S, Schimke M, Maya-Ramos L, Aroonsakool N, et al. Role for high-glucose-induced protein O-GlcNAcylation in stimulating cardiac fibroblast collagen synthesis. *AJP Cell Physiol* [Internet]. 2014;306(9):C794–804. Available from: <http://ajpcell.physiology.org/cgi/doi/10.1152/ajpcell.00251.2013>
10. Higgins CB, Zhang Y, Mayer AL, Fujiwara H, Stothard AI, Graham MJ, et al. Hepatocyte ALOXE3 is induced during adaptive fasting and enhances insulin sensitivity by activating hepatic PPAR γ . *JCI Insight* [Internet]. 2018;3(16). Available from: <https://doi.org/10.1172/jci.insight.120794>
11. Watanabe M, Houten SM, Matakci C, Christoffolete MA, Kim BW, Sato H, et al. Bile acids induce energy expenditure by promoting intracellular thyroid hormone activation. *Nature*. 2006;439(7075):484–9.
12. Marvyn PM, Bradley RM, Mardian EB, Marks KA, Duncan RE. Data on oxygen consumption rate, respiratory exchange ratio, and movement in C57BL/6J female mice on the third day of consuming a high-fat diet. *Data Br* [Internet]. 2016;7:472–5. Available from: <http://www.sciencedirect.com/science/article/pii/S2352340916300889>
13. Randle PJ. Regulatory interactions between lipids and carbohydrates: the glucose fatty acid cycle after 35 years - Randle - 1998 - Diabetes/Metabolism Research and Reviews - Wiley Online Library. *Diabetes-Metabolism Rev* [Internet]. 1998;283(August):263–83. Available from: [http://onlinelibrary.wiley.com/doi/10.1002/\(SICI\)1099-0895\(199812\)14:4%3C263::AID-DMR233%3E3.0.CO;2-C/abstract](http://onlinelibrary.wiley.com/doi/10.1002/(SICI)1099-0895(199812)14:4%3C263::AID-DMR233%3E3.0.CO;2-C/abstract)
14. Muoio DM, Newgard CB. Molecular and metabolic mechanisms of insulin resistance and β -cell failure in type 2 diabetes. *Nat Rev Mol Cell Biol* [Internet]. 2008 Mar 1;9:193. Available from: <http://dx.doi.org/10.1038/nrm2327>
15. Liu Q, Bengmark S, Qu S. The role of hepatic fat accumulation in pathogenesis of non-alcoholic fatty liver disease (NAFLD). *Lipids Health Dis* [Internet]. 2010;9(1):42. Available from: <https://doi.org/10.1186/1476-511X-9-42>
16. Utzschneider KM, Kahn SE. The Role of Insulin Resistance in Nonalcoholic Fatty Liver Disease. *J Clin Endocrinol Metab* [Internet]. 2006 Dec 1;91(12):4753–61. Available from: <http://dx.doi.org/10.1210/jc.2006-0587>

17. Yang Z, Ming XF. Functions of arginase isoforms in macrophage inflammatory responses: Impact on cardiovascular diseases and metabolic disorders. *Front Immunol*. 2014;5(OCT):1–10.
18. Xiong Y, Yepuri G, Forbitech M, Yu Y, Montani J, Xiong Y, et al. regulation of MTOR and PRKAA / AMPK signaling in advanced atherosclerosis ARG2 impairs endothelial autophagy through regulation of MTOR and PRKAA / AMPK signaling in advanced atherosclerosis. 2014;8627(September 2016).
19. Gómez-Herreros F, Zagnoli-Vieira G, Ntai I, Martínez-Macías MI, Anderson RM, Herrero-Ruiz A, et al. TDP2 suppresses chromosomal translocations induced by DNA topoisomerase II during gene transcription. *Nat Commun* [Internet]. 2017;8(1):233. Available from: <https://doi.org/10.1038/s41467-017-00307-y>
20. Wang P, Wander CM, Yuan C-X, Bereman MS, Cohen TJ. Acetylation-induced TDP-43 pathology is suppressed by an HSF1-dependent chaperone program. *Nat Commun* [Internet]. 2017;8(1):82. Available from: <https://doi.org/10.1038/s41467-017-00088-4>
21. DeBosch BJ, Heitmeier MR, Mayer AL, Higgins CB, Crowley JR, Kraft TE, et al. Trehalose inhibits solute carrier 2A (SLC2A) proteins to induce autophagy and prevent hepatic steatosis. *Sci Signal* [Internet]. 2016 Feb 23;9(416):ra21-ra21. Available from: <http://stke.sciencemag.org/content/9/416/ra21.abstract>
22. Wing RR, Lang W, Wadden TA, Safford M, Knowler WC, Bertoni AG, et al. Benefits of Modest Weight Loss in Improving Cardiovascular Risk Factors in Overweight and Obese Individuals With Type 2 Diabetes. *Diabetes Care* [Internet]. 2011 Jul 17;34(7):1481–6. Available from: <http://www.ncbi.nlm.nih.gov/pmc/articles/PMC3120182/>
23. Pernow J, Jung C. Arginase as a potential target in the treatment of cardiovascular disease: Reversal of arginine steal? *Cardiovasc Res*. 2013;98(3):334–43.
24. Longo VD, Mattson MP. Fasting: Molecular mechanisms and clinical applications. *Cell Metab* [Internet]. 2014;19(2):181–92. Available from: <http://dx.doi.org/10.1016/j.cmet.2013.12.008>
25. Longo VD, Panda S. Fasting, Circadian Rhythms, and Time-Restricted Feeding in Healthy Lifespan. *Cell Metab* [Internet]. 2016;23(6):1048–59. Available from: <http://dx.doi.org/10.1016/j.cmet.2016.06.001>
26. Mattson MP, Wan R. Beneficial effects of intermittent fasting and caloric restriction on the cardiovascular and cerebrovascular systems. *J Nutr Biochem*. 2005;16(3):129–37.
27. Iynedjian PB, Pilot PR, Nospikel T, Milburn JL, Quaade C, Hughes S, et al. Differential expression and regulation of the glucokinase gene in liver and islets of Langerhans. *Proc Natl Acad Sci U S A*. 1989;86(20):7838–42.

REVIEWERS' COMMENTS:

Reviewer #1 (Remarks to the Author):

N/A

Reviewer #3 (Remarks to the Author):

The authors have addressed our concerns pretty extensively and have performed some significant new experiments to complement their data.

Reviewer #4 (Remarks to the Author):

When I read this excellent paper by de Bosch and colleagues, I was reminded of the fable of the blindfolded Pygmies examining the elephant. They each had a different part of the elephant and differed in their description on this basis. I came to this review as someone who has been studying the arginases for many years and with a specific vision of them. It was jarring to find none of this legacy reflected in the paper. The arginases are ancient; they are highly conserved in evolution with some runs of amino acids unchanged throughout a billion years; the so-called Arg2 is in fact the ancestral gene involved in ornithine biosynthesis; the so-called Arg1 having evolved in amphibians to dispose of urea; Arg2 is distinguishable from Arg1 in some enzymatic properties; Arg2, but not Arg1 is located in the mitochondrion (more about that in a moment); and arg2 has some enzymatic function in Arg1 deficiency in man and animals and is highly "induced" in this deficiency, especially in man. Finally, Arg2 function and location has never been fully pinned down, although as the authors indicate there has been nibbling at the periphery of this with implications that it may have a role in macrophage function, and in diabetes, obesity and hypertension.

Using the approach of discovery science, the authors stumbled on the fact that Arg2 may be involved in a very central way in the regulation of energy metabolism in glucose deprivation and many accessory functions. In this paper they sought to prove the centrality of this regulatory pathway and in my view they have. It was very exciting for me to read these data which demonstrated a primary non-enzymatic function for this enzyme, although we were treated to the pathway it uses, but not

yet how this is accomplished and why it was chosen for this function. I was specifically asked by the editor to determine whether or not the authors addressed the original critique of the first set of reviewers. I believe that they were successful in doing this. Each of the phenomenally detailed criticisms were addressed. In some cases they performed an entirely new experiments to address the points and in others they defended their conclusions. In coming to this conclusion I can honestly say that I have never worked so hard to read and understand a paper and to write a review. The data were voluminous, reading the paper required jumping from text to figures, to the supplementary material and back. I can only come back to my original conclusion, this paper and its conclusions are excellent and well-supported and the paper is heuristic and a “game-changer”. It is worthy for publication in as prestigious an outlet such as Nature Communications.

Does this mean that the paper is perfect? Of course not. However, I generally side with those who see the glass as half full rather than half empty. Not every experiment will prove itself in future studies, but the paradigm of the future has been established and that is important. I have to agree with both previous reviewers that despite the claims for statistical significance some of the absolute differences in the studies are so small that you have to wonder about their biological importance. This is true at least to figure 1I and some of the values in figure 2. I never could figure out what was being shown in figure 2C, just as example.

One might wonder about my first paragraph and its relevance to this review. It underscores issues that a urea cycle person would latch onto where an energy metabolism person might gloss over. For example, some of the data from the cell culture experiments are artificial. For example, ureagenesis in the cell culture studies were enhanced by the transfection of Arg2. This probably would not occur in vivo, since arginine levels in liver are virtually undetectable in normal animals. They get ureagenesis because there is arginine in the medium. They show this themselves in the mouse studies where arginine is far lower than any other amino acid. If they noticed this they don't comment on it. The location of the transfected arginase is never shown in any experiment. Is it in the mitochondrion, somewhere in transit or in a biological unnatural environment. The authors have chosen to express various changes as a percent of baseline. This works for most things but it obscures the fact that even with 20-fold elevation, the Arg2 is much lower than Arg1 in hepatocytes and presumably the cells. It, among other things explains why there is no augmentation of ureagenesis. The arginases operate at the K_m at physiological concentrations, again not so important because we really don't think that Arg2 in this system is operating as an enzyme but rather in some “moonlighting role”.

In the larger picture, these disparities have not bothered the target audience of metabolism specialists and none actually detract from their conclusions. I would like to see this work see the light of day as quickly as possible.

RESPONSES TO REVIEWER COMMENTS:

Reviewer #1 (Remarks to the Author):

N/A

Response: Thank you.

Reviewer #3 (Remarks to the Author):

The authors have addressed our concerns pretty extensively and have performed some significant new experiments to complement their data.

Response: Thank you.

Reviewer #4 (Remarks to the Author):

When I read this excellent paper by de Bosch and colleagues, I was reminded of the fable of the blindfolded Pygmies examining the elephant. They each had a different part of the elephant and differed in their description on this basis. I came to this review as someone who has been studying the arginases for many years and with a specific vision of them. It was jarring to find none of this legacy reflected in the paper. The arginases are ancient; they are highly conserved in evolution with some runs of amino acids unchanged throughout a billion years; the so-called Arg2 is in fact the ancestral gene involved in ornithine biosynthesis; the so-called Arg1 having evolved in amphibians to dispose of urea; Arg2 is distinguishable from Arg1 in some enzymatic properties; Arg2, but not Arg1 is located in the mitochondrion (more about that in a moment); and arg2 has some enzymatic function in Arg1 deficiency in man and animals and is highly "induced" in this deficiency, especially in man. Finally, Arg2 function and location has never been fully pinned down, although as the authors indicate there has been nibbling at the periphery of this with implications that it may have a role in macrophage function, and in diabetes, obesity and hypertension.

Using the approach of discovery science, the authors stumbled on the fact that Arg2 may be involved in a very central way in the regulation of energy metabolism in glucose deprivation and many accessory functions. In this paper they sought to prove the centrality of this regulatory pathway and in my view they have. It was very exciting for me to read these data which demonstrated a primary non-enzymatic function for this enzyme, although we were treated to the pathway it uses, but not yet how this is accomplished and why it was chosen for this function. I was specifically asked by the editor to determine whether or not the authors addressed the original critique of the first set of reviewers. I believe that they were successful in doing this. Each of the phenomenally detailed criticisms were addressed. In some cases they performed an entirely new experiment to address the points and in others they defended their conclusions. In coming to this conclusion I can honestly say that I have never worked so hard to read and understand a paper and to write a review. The data were voluminous, reading the paper required jumping from text to figures, to the supplementary material and back. I can only come back to my original conclusion, this paper and its conclusions are excellent and well-supported and the paper is heuristic and a "game-changer". It is worthy for publication in as prestigious an outlet such as Nature Communications.

Does this mean that the paper is perfect? Of course not. However, I generally side with those who see the glass as half full rather than half empty. Not every experiment will prove itself in future studies, but the paradigm of the future has been established and that is important. I have to agree with both previous reviewers that despite the claims for statistical significance some of the absolute differences in the studies are so small that you have to wonder about their biological importance. This is true at least to figure 1 and some of the values in figure 2. I never could figure out what was being shown in figure 2C, just as example.

One might wonder about my first paragraph and its relevance to this review. It underscores issues that a urea cycle person would latch onto where an energy metabolism person might gloss over. For example, some of the data from

the cell culture experiments are artificial. For example, ureagenesis in the cell culture studies were enhanced by the transfection of Arg2. This probably would not occur in vivo, since arginine levels in liver are virtually undetectable in normal animals. They get ureagenesis because there is arginine in the medium. They show this themselves in the mouse studies where arginine is far lower than any other amino acid. If they noticed this they don't comment on it. The location of the transfected arginase is never shown in any experiment. Is it in the mitochondrion, somewhere in transit or in a biological unnatural environment. The authors have chosen to express various changes as a percent of baseline. This works for most things but it obscures the fact that even with 20-fold elevation, the Arg2 is much lower than Arg1 in hepatocytes and presumably the cells. It, among other things explains why there is no augmentation of ureagenesis. The arginases operate at the Km at physiological concentrations, again not so important because we really don't think that Arg2 in this system is operating as an enzyme but rather in some "moonlighting role".

In the larger picture, these disparities have not bothered the target audience of metabolism specialists and none actually detract from their conclusions. I would like to see this work see the light of day as quickly as possible.

-Stephen D. Cedarbaum

Response: Thank you for the comments, Dr. Cedarbaum. In response, we added to this manuscript's Discussion to take a broader view of mammalian arginase function. We incorporated a broader literature and thus expect that the appeal is in line with the broad interdisciplinary readership of *Nature Communications*. Also, we discussed the modest absolute phenotypes for some of our readouts and placed the data in context of its relevance to human patients with metabolic disease. An excerpt pertaining to these issues is demonstrated below and in the revised Discussion section:

"Arginases are ancient, highly conserved enzymes (36, 37), the purpose of which is long considered to be conversion of L-arginine to urea and ornithine (36–43). More recently, these enzymes have been implicated in the biosynthesis of polyamines, and in proline, glutamate, creatine, and agmatine biosynthesis as well (38, 40, 42, 44). The two vertebrate arginases, type I and type II evolved separately from an early gene duplication event (36), which ultimately manifests as both distinct and overlapping enzymatic functions, which yet remain to be fully appreciated (42). We demonstrate here that Arg2 is more profoundly induced during fasting in liver versus Arg1. This opens the possibility now that the arginases regulate fasting biology in the liver. Although the total abundance of Arg1 is much greater than that of Arg2 in the basal setting (42), the data here suggest that Arg1 and Arg2 are in fact differentially regulated during fasting, and may thus serve divergent functions as part of the physiological fasting response. The fundamental biological purpose of hepatocyte Arg2 regulation during fed and fasting states, and during health and disease remains an intriguing area of further exploration.

...Although it must be noted that some of the individual metabolic improvements observed upon AAV8-Arg2 expression were modest. For example, we observed 20-25% improvements in glucose and insulin tolerance in HFrD-fed AAV8-Arg2 and *db/db* x AAV8-Arg2 mice. However, it should also be noted that the magnitude of changes in glucose and insulin tolerance here mirrors that observed in humans subjected to short-term intermittent fasting or caloric restriction, which results in 11-30% improvements in HOMA-IR, and 11-40% improvement in fasting plasma insulin levels (7). Indeed even high-intensity, supervised diet and exercise training regimens over a 4-month period produced even more modest improvements in glucose tolerance, and only for the highest-intensity intervention groups (45)."